## METHOD

# Lineage abundance estimation for SARS-CoV-2 in wastewater using transcriptome quantification techniques

Jasmijn A. Baaijens[1,2*†] , Alessandro Zulli[3†], Isabel M. Ott[4†], Ioanna Nika[2], Mart J. van der Lugt[2], Mary E. Petrone[4], Tara Alpert[4], Joseph R. Fauver[4,5], Chaney C. Kalinich[4], Chantal B. F. Vogels[4], Mallery I. Breban[4], Claire Duvallet[6], Kyle A. McElroy[6], Newsha Ghaeli[6], Maxim Imakaev[6], Malaika F. Mckenzie-Bennett[7], Keith Robison[7], Alex Plocik[7], Rebecca Schilling[7], Martha Pierson[7], Rebecca Littlefield[7], Michelle L. Spencer[7], Birgitte B. Simen[7], Yale SARS-CoV-2 Genomic Surveillance Initiative, William P. Hanage[8], Nathan D. Grubaugh[4,9†], Jordan Peccia[3†] and Michael Baym[1†]

†Jasmijn A. Baaijens, Alessandro Zulli, and Isabel M. Ott share equal contributions.

†Nathan D. Grubaugh, Jordan Peccia, and Michael Baym co-senior authorship.

*Correspondence:
j.a.baaijens@tudelft.nl

[2] Department of Intelligent Systems, Delft University of Technology, Delft, Netherlands Full list of author information is available at the end of the article

## Abstract

Effectively monitoring the spread of SARS-CoV-2 mutants is essential to efforts to counter the ongoing pandemic. Predicting lineage abundance from wastewater, however, is technically challenging. We show that by sequencing SARS-CoV-2 RNA in wastewater and applying algorithms initially used for transcriptome quantification, we can estimate lineage abundance in wastewater samples. We find high variability in signal among individual samples, but the overall trends match those observed from sequencing clinical samples. Thus, while clinical sequencing remains a more sensitive technique for population surveillance, wastewater sequencing can be used to monitor trends in mutant prevalence in situations where clinical sequencing is unavailable.

## Background

As the SARS-CoV-2 pandemic continues, the virus is evolving in real time, challenging existing control measures. Increased infectivity, and potentially increased morbidity and immune evasion, have been observed in emerging lineages [1, 2]. A SARS-CoV-2 lineage is a group of closely related viruses with a common ancestor defined by key mutations, some of which are likely to be selective adaptations of the SARS-CoV-2 virus [1]. Some lineages, or groups of lineages, with public health interest are then classified as variants. For optimal pandemic management and public health strategies, we need first to be able to observe which lineages are present where, and critically how the relative abundance of these lineages is changing in the population.

Genomic surveillance of SARS-CoV-2 enables early detection and clinical investigation of emerging lineages [3]. In late 2020, the Centers for Disease Control and Prevention (CDC) designated specific viral lineages as *variants of interest* or *variants of concern* based on potential changes in detectability, transmissibility, disease severity, therapeutic efficacy, and/or ability to evade control by natural or vaccine-induced immune responses [4]. Initially, this designation included lineage B.1.1.7 (corresponding to WHO variant designation *Alpha*), B.1.351 (*Beta*), and P.1 (*Gamma*). In early 2021, CDC added two new variants to this list: B.1.427 and B.1.429 (*Epsilon*), both of which were first identified in California [5, 6]. B.1.617.2 and related sublineages (*Delta*) now pose a threat but were not observed at substantial rates in the United States until May 2021. Each lineage is identified by a set of potentially overlapping amino acid mutations, which can be identified by genome sequencing. This is typically done by sequencing remnant clinical samples used for diagnostics (e.g., nasal swabs), but as infected patients excrete high levels of SARS-CoV-2 RNA, lineage prevalences are potentially detectable from domestic wastewater.

Measuring the concentration of SARS-CoV-2 in domestic wastewater can be an efficient method for indicating infection dynamics in a population [7]. SARS-CoV-2 and fragments of its RNA genome are excreted by infected individuals through feces or urine [8] and collected in domestic wastewater. Viral RNA in wastewater can then be extracted and quantified via quantitative RT-qPCR. This approach has been used to measure SARS-CoV-2 abundance over time, across different regions [9] and wastewater RNA concentrations are correlated with COVID-19 case rates [10, 11]. The genomes of SARS-CoV-2 in wastewater can also be sequenced, which can then be used to identify mutations present in an entire community with respect to the reference genome [12, 13]. Genome analysis from wastewater sequencing is particularly challenging because of the low concentrations and poor, fragmented quality of RNA, and the presence of PCR-inhibiting compounds which can interfere with library preparation in wastewater. This typically yields poor-quality sequencing data where the sequencing depth is highly variable across the SARS-CoV-2 genome and overall genome coverage is often incomplete. Despite these challenges, recent work has shown that it is feasible to observe individual mutations in wastewater sequencing data [12, 13], and suggests the possibility of monitoring the abundance of lineages of concern or interest. Throughout the world, SARS-CoV-2 wastewater surveillance has been conducted for wastewater collection systems that serve populations ranging from 10,000 to greater than 100,000 people [14]. A method for the quantitative measurement of lineages in wastewater would provide a cost- and resource-efficient approach to population genome surveillance.

Given the urgency of wastewater surveillance during the ongoing pandemic, several tools are under development to tackle the problem of lineage (or variant) quantification from wastewater sequencing data. For example, Ellmen et al. define an optimization problem to combine individual mutation frequencies into lineage frequencies [15] and Karthikeyan et al. propose an algorithm based on a regression problem to minimize the edit distance between sequences and a reference [16]. Each of these approaches under development relies on the discovery and quantification of individual mutations using popular tools like V-pipe [17] or iVar [18], a process that is highly error-prone given the nature of wastewater sequencing data.

Here we introduce a technique to monitor for SARS-CoV-2 lineages in a population by sequencing directly from wastewater and predicting abundances via a computational approach previously used for RNA-seq transcript quantification. The main strength of this algorithm is that it does not rely on discovery and quantification of individual mutations and it is not affected by bias towards a single reference genome, since it uses a complete reference collection representative of all known SARS-CoV-2 lineages. We demonstrate the efficacy of our VLQ pipeline (Viral Lineage Quantification) on wastewater data collected from Connecticut between January and April 2021, during the third wave of the SARS-CoV-2 pandemic, and compare these predictions to clinically observed variant frequencies from the same geographic area and time period. We then show the generality of the approach by expanding our analysis to samples collected across the United States from late December 2020 to January 2021.

## Results

### Prediction of lineage abundance is computationally analogous to RNA transcript abundance estimation

SARS-CoV-2 RNA fragments in wastewater originate from different people carrying a SARS-CoV-2 infection, with potentially different viral lineages, pooled together into a single sample. After successfully extracting RNA from wastewater and sequencing SARS-CoV-2 genome fragments, the computational challenge is to assign reads to lineages and estimate relative abundance per lineage. This is analogous to RNA transcript quantification from RNA-Seq data, where the sequencing data consists of reads originating from different transcripts of a given gene, and the objective is to quantify the relative abundance per transcript (Fig. 1a).

RNA transcript quantification algorithms make use of the genome sequences without identifying specific mutations. This is a major advantage because identifying individual mutation frequencies from wastewater sequencing data is highly error-prone—any errors would be propagated into lineage abundance estimates. Moreover, the fact that RNA transcript quantification tools have already been in use for several years in the RNA-Seq community has resulted in well-developed, user-friendly software that can be applied almost immediately to the lineage abundance quantification problem.

It is important to note that the computational task of lineage abundance prediction from wastewater sequencing data is also highly similar to the viral haplotype reconstruction problem (commonly referred to as viral quasispecies assembly). There, the input consists of sequencing data from a single infection where within-host evolution has resulted in a so-called quasispecies: a cloud of closely related mutants that together comprise the infection [19]. While this input seems similar to wastewater sequencing data, the crucial difference is that wastewater samples suffer from factors such as inhibitory compounds, RNA degradations, low RNA concentrations, and frequent amplicon drop-out. Per consequence, regular viral haplotype reconstruction approaches (see Eliseev et al. [20] for an overview) are not well suited for this type of data—instead, we need an approach that is robust to the variability in wastewater sequencing data quality. Moreover, available viral haplotype assembly and quantification tools do not take into account a set of pre-defined lineages, hence the computational problem addressed is different from the lineage abundance quantification problem.

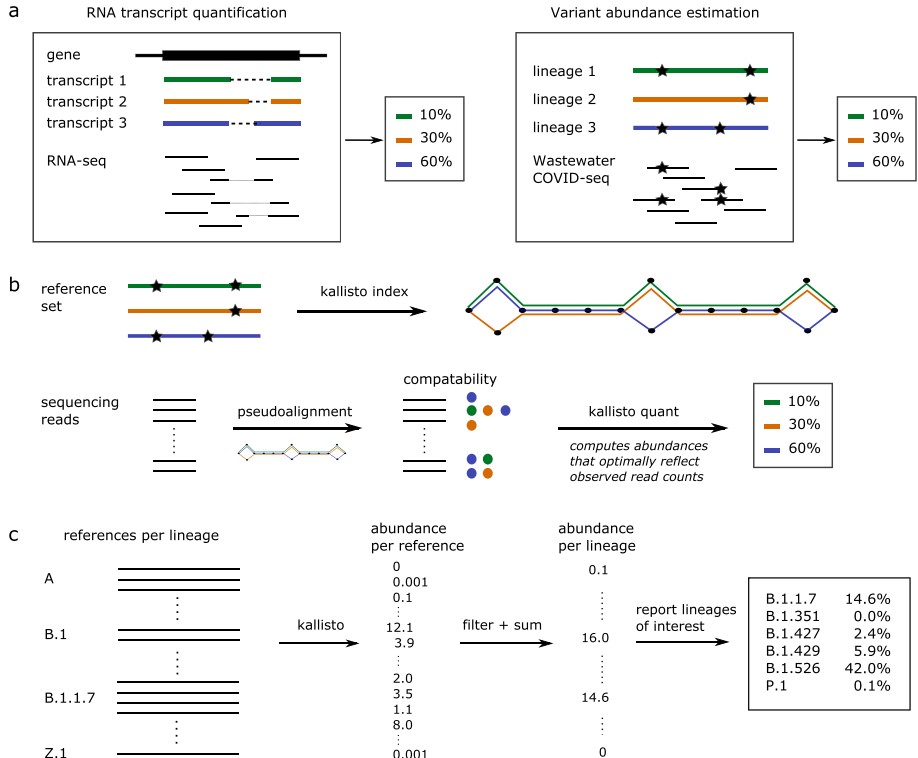

**Fig. 1** VLQ, a computational approach to lineage abundance estimation from wastewater sequencing data. **a** Computational similarity between RNA transcript quantification and lineage abundance estimation. **b** Key aspects of the kallisto algorithm in the context of lineage abundance estimation. **c** Our workflow uses multiple reference sequence per lineage to capture within-lineage variation. Applying kallisto (as in part b) results in abundance estimates per reference sequence. These abundances are filtered using a minimal abundance cutoff and subsequently summed per lineage to obtain abundance estimates per lineage. Finally, lineage abundances are reported

Here, we predict lineage abundances by applying kallisto [21]. This algorithm takes as input a set of reference sequences to be quantified: for RNA-Seq, these would be the different transcripts, but for wastewater sequencing data we provide it with a collection of SARS-CoV-2 genomes representative of the population. kallisto constructs an index from the reference sequences and subsequently matches sequencing reads to references, which allows it to estimate the abundance of each reference transcript provided (Fig. 1b). SARS-CoV-2 lineages are characterized by a combination of mutations, but additional variation is observed within lineages (Fig. S1, Additional file 1). We provide kallisto with a reference set consisting of multiple genomic sequences per lineage, capturing the mutations specific to this lineage as well as within-lineage variation (Fig. 1c). Our constructed reference set includes 1-17 genome sequences per lineage, with a total number of nearly 1500 sequences for the 881 unique SARS-CoV-2 lineages present in the GISAID [22] database at the time of download (9 March 2021). Including multiple sequences per lineage reduces biases related to within-lineage variation and potentially identifies any additional genomic signatures frequently seen in a given lineage. Finally, we filter out any predictions below a given percent abundance threshold to reduce noise and sum all predictions per lineage, which gives predicted

abundances per lineage (Fig. 1c). While in this study we used kallisto, we expect similar results with comparable tools such as salmon [23].

## kallisto predictions of lineage abundance are accurate on simulated data

To evaluate the accuracy of the predictions obtained through our VLQ pipeline, we created a collection of benchmarking datasets that resemble real wastewater samples. For each lineage designated as a variant of concern at the time of these experiments (B.1.1.7, B.1.351, B.1.427, B.1.429, P.1) we created a series of 33 benchmarks by simulating sequencing reads from a variant genome, as well as a collection of background (non-variant of concern/interest) sequences, such that the variant lineage abundance ranges from 0.05% to 100%. Analogously, we created a second series of benchmarks, simulating reads only from the Spike gene of each SARS-CoV-2 genome. We refer to the first set of benchmarks as "whole genome" and to the second set of benchmarks as "Spike-only." Finally, we performed these benchmarking experiments at different sequencing depths: 100× and 1000× coverage for the whole genome benchmarks, and 100×, 1000×, and 10,000× coverage for the Spike-only benchmarks (Table 1 and Fig. 2). In Additional file 1, Section 3.2, we repeat the whole genome experiments at 1000× coverage after randomly downsampling the simulated data per window of 400 bp, to simulate the effect of amplicon bias.

Predicting lineage abundance can be difficult when a lineage is present at very low frequency, because of the high degree of similarity between lineages. On our simulated datasets, where we know the true frequency of each lineage, we observe a background noise of 0.01–0.09% (Fig. S2, Additional file 1), meaning that some sequences are falsely predicted to be present at 0.01–0.09% abundance. These false positives are likely due to shared mutations or conserved sequences between lineages. The level of background noise tends to be higher for whole genome benchmarks than for Spike-only benchmarks, because the majority of defining lineage mutations are in the Spike gene (Fig. S1, Additional file 1 and Fig. 2). In both cases, we apply a threshold of 0.1% abundance to include a sequence in the lineage abundance computation and we only report the presence of a lineage exceeding this threshold to avoid false positives. For this reason, we only report results for benchmarks with a true variant lineage abundance of at least 0.1%. Note that this threshold applies to the overall sequence

**Table 1** Performance statistics per dataset. Results separated by a forward slash correspond to an abundance threshold of 0.1% and 1%, respectively

| Benchmark | FPR | FNR | Precision | Recall | Relative estimation error (%) |
|---|---|---|---|---|---|
| Whole genome 100× | 0.191/0.0 | 0.057/0.032 | 0.423/1.0 | 0.943/0.968 | 29.4/19.4 |
| Whole genome 1000× | 0.163/0 | 0.007/0.042 | 0.470/1.0 | 0.993/0.958 | 27.1/18.5 |
| Spike-only 100× | 0.121/0.003 | 0.107/0.074 | 0.508/0.978 | 0.893/0.926 | 26.3/15.8 |
| Spike-only 1000× | 0.041/0.003 | 0.043/0.042 | 0.753/0.978 | 0.957/0.958 | 17.3/14.0 |
| Spike-only 10,000× | 0.010/0 | 0.014/0.042 | 0.926/1.0 | 0.986/0.958 | 15.3/13.0 |

*FPR* false positive rate = # false positives/(# false positives + # true negatives), *FNR* false negative rate = # false negatives/(# false negatives + # true positives); relative estimation error reflects the average relative frequency estimation error across all true positives

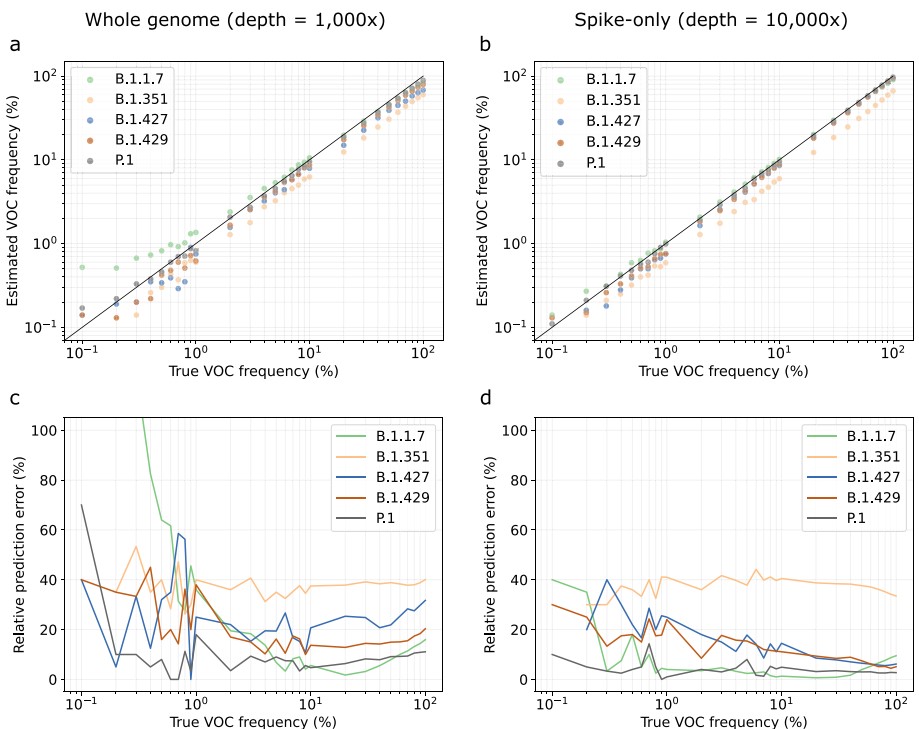

**Fig. 2** Benchmarking results on simulated wastewater sequencing data of a representative mixture of background sequences and a variant lineage (VOC; here: B.1.1.7, B.1.351, B.1.427, B.1.429, P.1). **a** Estimated lineage abundance (VOC frequency) versus true abundance on whole genome sequencing data with a depth of 1000×. **b** Estimated lineage abundance (VOC frequency) versus true abundance on Spike-only sequencing data with a depth of 10,000×. **c**, **d** Relative prediction error per lineage for the estimated frequencies presented in panels **a** and **b**, respectively. Relative prediction errors are defined as the absolute difference between true and estimated frequency, relative to the true frequency. VOC, variant of concern

abundance and not to individual mutations, since the stochasticity of wastewater sequencing causes the abundance of mutations within a lineage to vary significantly.

Figure 2a, b shows the predicted versus true frequencies per variant lineage for two of the benchmarks; additional results are shown in Fig. S3, Additional file 1. In general, lineage frequencies tend to be underestimated, in particular on whole genome data. This is another consequence of shared polymorphisms between closely related lineages: a fraction of the reads is assigned to other, locally identical genomes, leading to an underestimated lineage frequency. Additional benchmarking experiments show that this effect be explained by the presence of sequencing errors (Figs. S4 and S5, Additional file 1). We observe that the more divergent a lineage is in comparison with other lineages and the more unique polymorphisms are associated with it, the lower the number of false positives and the smaller the underestimation of lineage abundance. This explains why our predictions are most accurate for P.1, the most divergent lineage among the variants considered (Fig. S1, Additional file 1).

Figure 2c, d shows the relative frequency estimation error, defined as the absolute difference between true and estimated frequency, relative to the true frequency. In other words, this metric evaluates the prediction accuracy relative to the true frequency. We observe that relative frequency estimation errors are highest at low

frequencies—indeed, at low frequencies a small deviation of the predicted abundance in the absolute sense makes a large relative difference. Figure 2 shows that we can predict lineage frequencies of at least 1% at high accuracy, with relative frequency estimation errors which are relatively stable.

At low lineage frequencies, we also notice overestimation in the predicted lineage abundance. This is due to shared sequence between the variant lineage and the background lineages: in datasets where the variant is present at a low frequency, the background lineages must be present at relatively high frequencies. Any shared sequence between a background lineage and the variant will then lead to more reads being assigned to the variant, hence overestimation. This effect only applies when background lineages with shared sequence are more abundant than the variant lineage. In Fig. 2a, we can clearly see how P.1 and B.1.1.7 abundances are overestimated at low frequencies, while being near-perfect at higher frequencies (>1%); for other variants, we see that this effect compensates some of the underestimation, resulting in better estimates at low variant frequencies.

Among the variant lineages that we evaluate prediction accuracy for, lineages B.1.427 and B.1.429 are particularly difficult to predict individually. These lineages are highly similar and have the same characterizing mutations in the Spike gene [24]. Figure 2b, d shows that, despite this inherent difficulty, we can accurately predict abundance for these lineages. This highlights one of the main strengths of our approach: because we include multiple sequences per lineage in our reference set, we capture within-lineage variation that allows us to distinguish between highly similar lineages, even if there are no characteristic amino acid mutations to distinguish them by. Additional benchmarking experiments show that we can distinguish between highly similar sequences even at a mutation rate of only 0.01% (Fig. S6, Additional file 1).

We observe that lineage frequencies are consistently underestimated using our approach, except for slight overestimation of unusually divergent lineages (P.1, B.1.1.7) at low frequencies. Consistent prediction bias as observed in our experiments is unlikely to be an issue in differential analysis, but in single-point evaluations, it would be necessary to design a method to correct for these lineage-specific biases. However, our benchmarks are generated using a single genome sequence per variant lineage, while real data consists of a mixture of different sequences for the same variant. This may have resulted in a stronger underestimation on our benchmarks than would be seen on real data. More extensive benchmarking experiments will make it possible to learn the variant-specific biases more accurately and adjust predictions accordingly.

To evaluate false positive and false negative predictions, we computed for each experiment the overall false positive rate (FPR), false negative rate (FNR), precision, and recall (Table 1). Here, a false positive means that we falsely predict a variant lineage to be present (i.e., it is not present in the simulated sequencing data), and a false negative means that we predict an abundance of 0 for a variant lineage that is present in the simulated sequencing data. We calculated these statistics for minimal lineage abundance thresholds of 0.1% and 1%. Increasing the minimal abundance thresholds reduces the false positive rate but increases the false negative rate. We generally observe more false negatives for datasets of lower coverage, because low-frequency lineages become harder to detect. In terms of precision and recall, we note that, unsurprisingly, increasing the number of reads (either

by amplifying a larger region or by increased sequencing depth) leads to better results. If a lineage is uniquely defined by mutations on Spike, then sequencing depth is preferred over breadth, but if a variant lineage is (nearly) identical to other lineages on Spike, e.g., B.1.427/B.1.429, then whole genome sequencing is preferable.

While for this study we primarily used kallisto [21], we also evaluated performance for other transcript quantification tools. First, we evaluated the software package salmon [23], which takes a slightly different algorithmic approach to the same problem (Fig. S7, Additional file 1), and found predictions were highly similar to those obtained with kallisto, the main difference being that salmon is slightly more conservative: it achieves higher precision (fewer false positives), at the expense of lower recall (more false negatives). Second, we evaluated RSEM [25] and Iso-EM2 [26], both of which are EM-based algorithms based on regular read-to-transcriptome alignment [27]. While RSEM consistently overestimates lineage abundance, Iso-EM2 consistently underestimates abundance (Table S1, Additional file 1). Overall, kallisto performs best among this selection of popular transcript quantification algorithms. Although salmon tends to miss lineages at very low frequencies, one potential advantage is that this method may also be applied to long reads (in alignment-based mode), while kallisto usage is limited to short reads.

### Observed PCR values of wastewater correspond to genome coverage

We obtained primary sewage sludge samples from the wastewater treatment plant serving New Haven, CT, USA, every 2 days between January 1, 2021, and April 27, 2021 (59 samples). The observed SARS-CoV-2 RNA levels in these samples follow the same trend as COVID-19 case rates in the same geographic region (Fig. 3a, similarly found in [11]). We generated 400nt tiled amplicons encompassing the SARS-CoV-2 genome and performed genome sequencing (see the "Methods" section). The resulting sequencing data varied widely in terms of the number of reads and genome coverage (Fig. S8, Additional file 1), with low Ct values (high SARS-CoV-2 RNA concentrations) generally leading to higher genome coverage (Fig. 3b). For these datasets, Ct values < 31 yield at least 60% genome coverage; samples with a Ct value < 34 and at least 0.5M reads aligned reach a genome coverage of at least 40%.

To evaluate the impact of genome coverage on lineage abundance predictions, we subsampled a dataset with maximal coverage (99% of the SARS-CoV-2 genome with >20× coverage, 1.9M paired-end reads) to obtain datasets with reduced genome coverage by randomly selecting 20%, 40%, 60%, and 80% of amplicons, respectively, each of which we repeated 100 times. Figure 3c shows the resulting abundance predictions for B.1.1.7 per coverage value. We observe that the median predicted abundance is close to the predicted abundance at full coverage (dashed line in Fig. 3c) for all coverage values; however, variance is much larger in datasets with low coverage compared to datasets with high coverage, consistent with statistical predictions and prior work [18]. This indicates that datasets with low genome coverage can still result in accurate abundances, but the predictions are less reliable.

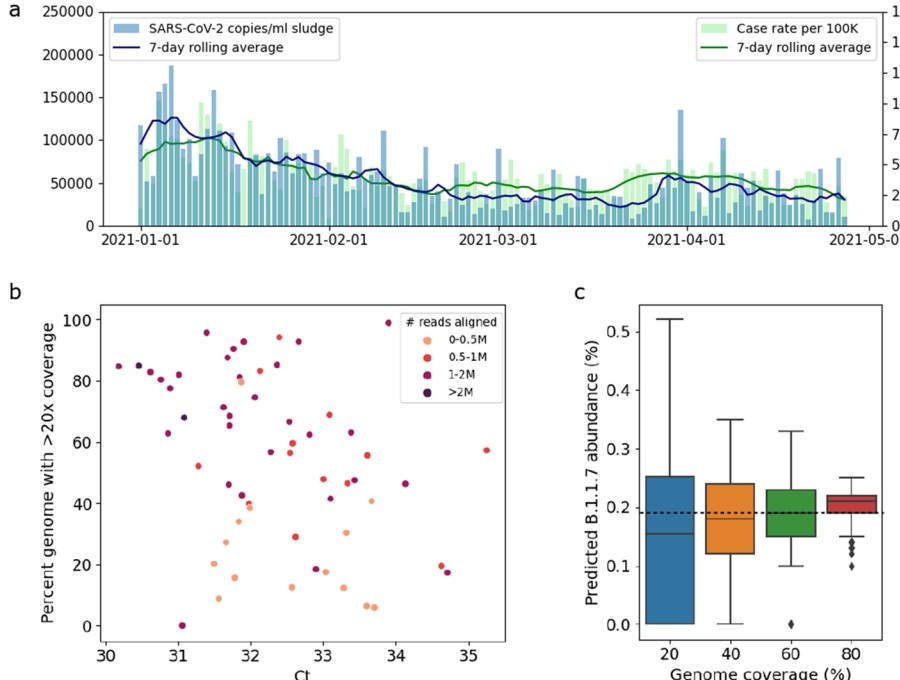

**Fig. 3** **a** RNA levels in wastewater (copies/ml sludge, displayed on the left vertical axis) follow the same trend as COVID-19 case rates (cases per 100K people, displayed on the right vertical axis). **b** Percent genome with >20× coverage versus sludge Ct values. **c** Impact of genome coverage on predicted B.1.1.7 abundance for random subsamples of a sludge sample with full genome coverage. The horizontal dotted line indicated the predicted B.1.1.7 abundance for the full sample (99% genome coverage)

## Wastewater abundances of B.1.1.7 and B.1.526 in Connecticut broadly correspond to clinically observed frequencies

We applied our abundance prediction pipeline to the series of wastewater sequencing datasets described above. Figure 4 shows the resulting predictions for lineages B.1.1.7 and B.1.526. There is a clear trend in B.1.1.7 abundance emerging in early February 2021, increasing in abundance through mid-April 2021, while the abundance of other variant lineages is relatively stable over time (see also Fig. S9, Additional file 1).

We then compared our wastewater abundance predictions to lineage frequency estimates from data generated by sequencing remnant clinical diagnostic samples (mostly nasal swabs) in New Haven County, CT (Fig. 4). We observe that B.1.1.7 abundances predicted from wastewater are underestimated compared to the clinical abundance data. Based on our benchmarking experiments on whole genome data (Fig. 2c) we expect frequencies to be underestimated by 5-40% (relative to the actual frequency), with stronger underestimation for lower frequencies. This is consistent with what we see in Fig. 4: the increase (and subsequent decrease) of B.1.1.7 abundance in wastewater is stronger than in clinical data because of this bias, while wastewater abundance predictions are very close to clinical predictions at frequencies of 60–70%. For B.1.526, both clinical and wastewater abundance is relatively stable over time. Figure 4 suggests that the detection threshold for these variant lineages is around 10%: for clinical frequencies below 10%, the wastewater-based predictions tend to be underestimated or lacking, while above this threshold we see a clear signal, most strongly for B.1.526. In theory, predictions will be

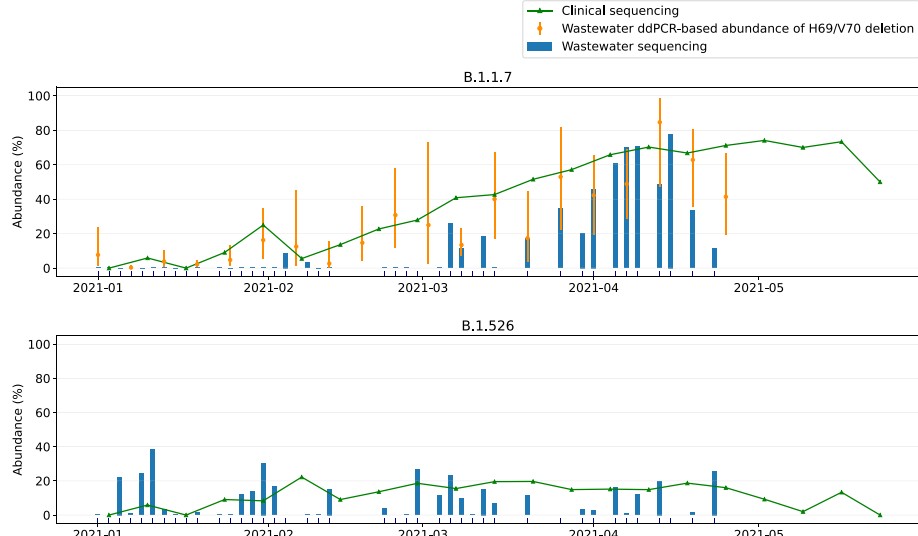

**Fig. 4** Wastewater versus clinical abundance estimates for B.1.1.7 and B.1.526 in New Haven from early January 2021 to late April 2021. Dates of clinical sampling correspond to the date of specimen collection. In addition, ddPCR-based abundance estimates of lineages with the H69/V70 deletion (likely B.1.1.7) are shown for wastewater samples taken every six days. Confidence intervals are computed from ddPCR confidence intervals for measured copied of the H69/V70 deletion and measured copies of wild type present (see the "Methods" section). Vertical dashes on the *x*-axis indicate timepoints where wastewater sequencing data was obtained that passed the filtering criteria (Ct value < 31 or Ct < 34 with at least 0.5M reads aligned)

closer to clinical abundances when sequencing only the Spike gene instead of the whole SARS-CoV-2 genome (Fig. 2). In practice, however, using only reads aligning to the Spike gene captures too little information due to incomplete genome coverage and amplification bias.

Kallisto offers a bootstrapping feature, through which the sequencing data is resampled at least 100 times and lineage abundances are predicted for each of these resampled datasets. The resulting predictions can subsequently be analyzed to obtain confidence intervals for the predicted abundance on the original dataset. For the New Haven sludge samples discussed here, we obtained narrow confidence intervals (upper and lower errors <1% abundance), suggesting that predictions are very consistent (Fig. S9, Additional file 1). However, this type of analysis captures only computational noise, and not technical noise (e.g., sampling bias). The fact that our predictions are more variable than the clinical data while confidence intervals from bootstrapping are narrow suggests that wastewater sequencing data is highly stochastic and not always representative of the infections in the population.

To complement the clinical data of lineage frequencies, we have analyzed our wastewater samples using ddPCR to estimate the prevalence of the B.1.1.7 lineage based on the presence of the characteristic H69/V70 deletion in the Spike gene (Fig. 4). We observe that ddPCR-based abundance estimates are highly variable. While at some time points the ddPCR-based abundace estimates are very close to the predictions obtained with kallisto, at other time points these estimates are quite different. We investigated these specific instances by aligning the sequencing reads to the SARS-CoV-2 reference genome (MN908947.3) and analyzing the alignments across the Spike deletion at amino acid positions 69–70 (as targeted by ddPCR). We observed that in these sequencing data

sets, this specific region is not covered by sequencing reads, thus confirming our earlier observation that wastewater sequencing data is highly stochastic.

**Wastewater abundances for different lineages across the US match expected patterns**
The quality of datasets obtained through wastewater sequencing varies widely. RNA levels and Ct values fluctuate with the number of infections in a sewershed, which impact overall genome coverage and prediction accuracy. Other factors such as sampling approach, PCR inhibition, and amplification bias add to this variability. To validate our approach for general use, we applied our pipeline to predict lineage abundance on a diverse collection of composite influent wastewater samples obtained from 25 treatment plants across the United States between late December 2020 and late January 2021 (Fig. 5). These samples were sequenced in-house and had slightly higher average Ct values than the sludge samples analyzed above (33.1 vs 32.3), and reduced genome coverage compared to sludge samples. Nevertheless, the same quality filtering parameters apply: Ct < 31 or Ct < 34 with at least 0.5M reads aligned to select for samples with high coverage (Fig. S10, Additional file 1).

After this filtering step, we predicted lineage abundance for the 30 remaining datasets (corresponding to 16 different locations across 8 states). Figure 5 shows the predictions for lineages B.1.1.7, B.1.427, B.1.429, and B.1.526, along with the clinical lineage frequencies in the corresponding state (calculated from GISAID in the 7-day window centered at the wastewater sampling date). Although data uploaded to GISAID has its own biases and individual towns do not necessarily reflect state-wide lineage abundances, this is the only statistic we can compare against across all states. We observe that, while individual samples are unreliable, the predicted lineage abundances match expected patterns across the US from the times of sampling: B.1.1.7 was predicted most abundantly in Florida; B.1.427 and B.1.429 were primarily found in California, and B.1.526 was predicted most abundantly in New York and Connecticut. Other lineages (B.1.351, P.1) were not observed in GISAID for these states at the time of sampling and our predictions for these lineages agree: B.1.351 was predicted to be present at very low frequency in 4 samples and absent in all other samples; P.1 was predicted present in a single dataset at 1% abundance and absent in all others (Fig. S11, Additional file 1). Although these predictions may be false positives, at the time P1 was thought to be likely at such a low prevalence that these cases were not picked up by the sequencing efforts in place.

**Discussion**
Our results show that methods for RNA transcript quantification can be applied to wastewater sequencing data to obtain consistent and relevant lineage abundance estimates. This technique can be readily applied to a wide range of data types, from Spike-only amplicon sequencing to whole genome sequencing; it is not appreciably more difficult than setting up a "reference set" of potential lineages and running existing tools on whatever the sequencing data happens to be. While this reference set approach allows easy updating as new lineages appear, it also means that this approach cannot be used to detect new lineages, but only to near-optimally impute the mixture of known lineages most likely responsible for the observed data. If a new lineage is present in the sequencing data, it would be quantified as the lineage with the most similar sequence

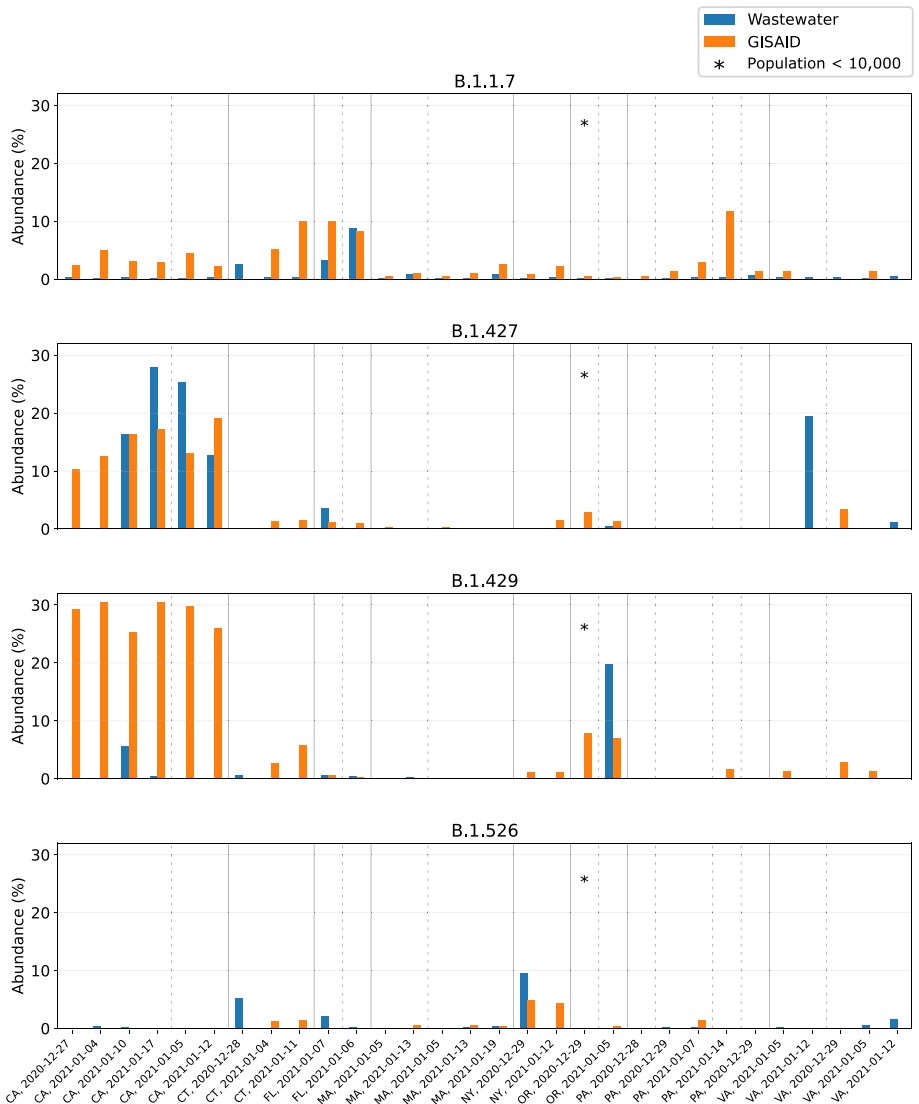

**Fig. 5** Wastewater versus GISAID abundance estimates for B.1.1.7, B.1.427, B.1.429, and B.1.526 at 16 locations across 8 states of the US. Samples were collected between late December 2020 and late January 2021; the sampling date and location are indicated on the horizontal axis. Samples are sorted by location, with different locations separated by a dotted line and different states separated by a solid line

(smallest edit distance), which is likely the most recent ancestor. Further, the approach may be readily extended to the detection and quantification of other pathogen lineages present in wastewater.

Detecting lineages in this fashion appears to be near-optimal on simulated data, with a detection limit under 1%, though when handling real data which is subject to multiple factors that can alter the quality of the RNA and the resulting signal, this may be closer to 10%. For real wastewater sequencing data, we observed that for clinical frequencies below 10% the wastewater-based predictions are often underestimated or lacking, while above this threshold we see a clear signal. Wastewater abundances generally follow the abundance trends seen in clinical data, though with sufficient noise that individual timepoints should not be considered reliable abundance estimates. Predicting lineage

abundance from wastewater sequencing is challenging when viral titers are low, as a result of low prevalence of SARS-CoV-2, but in high-prevalence regions, this approach can be extremely effective.

When comparing wastewater abundances to population-level clinical frequencies of the observed variant lineages, there are three distinct potential sources of errors, all of which are conflated in the accuracy of the final estimate: (*1*) how well clinical frequencies match the rates in the population as a whole, (*2*) how representative the RNA in a given wastewater is of the infections in the population as a whole, and (*3*) how accurately the predicted lineage abundances from sequencing represent what is in the wastewater sample itself. Clinical frequencies and the GISAID data used for the national data above have strong biases and so are not themselves ground truth. Individual wastewater samples can be unreliable: the catchment and composition of an individual wastewater sample can include hospital or industrial inputs and is not necessarily representative of the population, and low infection levels, inhibitory compounds, and degradation of RNA can result in higher Ct values and associated genome coverage.

Amplicon-based sequencing, as employed here, can also bias the detection of within-sample lineage frequencies if there are mismatches to the primer sequences. One of the strengths of using transcriptome analysis tools on such data is that we only analyze lineage abundance: as long as the amplicon biases even out across the genome, there will be little impact on prediction accuracy (Fig. S12, Additional file 1). However, the main limitation of this approach is that any amplification bias goes unnoticed, as we do not consider sequencing depth or mutation frequencies at individual sites. Further, the potential that different lineages have different viral shedding in waste is not taken into account, nor are differences in shedding rates for vaccine breakthrough cases. The data here is consistent with high-coverage sequencing being well representative of the sample and the computation faithfully reconstructing it, therefore we believe the primary source of observed noise is the underlying noisiness correspondence between a given wastewater sample, the population as a whole, and the clinical cases.

As part of our benchmarking experiments, we have attempted to incorporate the variability in sequencing depth (amplicon bias) in our simulated data and we have investigated the impact of sequencing errors on prediction accuracy. Both factors were shown to have little impact on the accuracy of our VLQ pipeline, provided that the sequencing error rate is within the expected range for Illumina sequencing (0.1–1%). It would be interesting to explore the use of long-read sequencing for wastewater analysis; in practice, however, RNA fragments may be too short to allow for high-throughput long-read sequencing due to RNA degradation in wastewater. Future approaches may also combine viral haplotype assembly methods (e.g., CliqueSNV [28], PredictHaplo [29], or SAVAGE [30]) with lineage quantification techniques like VLQ, to enable discovery of novel sequences along with quantification of known lineages. Due to the challenging nature of wastewater sequencing data, including RNA degradation and amplicon drop-out (see above), viral haplotype assemblers will struggle to reconstruct full-length sequences—instead, we expect assemblies to remain fragmented. The main challenge of working with these assemblies will be to assign the resulting fragments to lineages, similar to the lineage quantification task addressed here, with the additional possibility of assigning fragments to an unknown lineage. Besides exploring such combined approaches, it

would we valuable to generate real benchmarking data by creating artificial mixtures of lineages at pre-determined proportions, and evaluate robustness of the sequencing and computational steps.

The current results do offer lessons for future development of wastewater sequencing methods. Since sequencing a smaller genomic region allows for sequencing at higher depth at the same cost, we evaluated prediction accuracy on Spike-only benchmarking data. The Spike gene is a particularly variable region in the SARS-CoV-2 genome, hence it is highly informative for the purpose of lineage quantification. In fact, with perfect coverage (simulated data) Spike-only sequencing gives better predictions than whole genome sequencing. More broadly, as the lineage-discriminating mutations are restricted to a subset of the amplicons in tiled sequencing, a strategy focusing sequencing depth proportionally to the potentially informativity of the tiles would likely yield the more accurate predictions per sequence read. In practice, whole genome sequencing is often necessary to obtain enough information versus restricting to Spike. Differences in sequencing protocols, as well as differences in population size and sampling techniques, make it challenging to compare data between locations. Another important lesson is the improved coverage with higher virus RNA concentrations (lower PCR Ct values). While Ct is largely controlled by the infection rate in a community and excretion into wastewater, sample concentration and PCR inhibition removal approaches can be deployed to lower the Ct value and improve coverage in most samples.

In applying transcriptome quantification tools to lineage prediction, reference set composition is central. We were able to reduce lineage-specific biases substantially by including multiple reference sequences per lineage, thus capturing within-lineage variation to identify lineages with highly similar genomes. Preliminary experiments (data not shown) comparing a US-specific reference set to a global reference set showed that, unsurprisingly, the US-specific reference set gives more accurate predictions for benchmarks with sequences from US origin. This suggests that using a state- or county-specific reference set construction could further improve results and that identification of lineages is likely being aided by the presence of hitchhiker mutations which are locally over-represented or non-defining.

## Conclusions

We present VLQ, a computational approach for estimating the percent abundances of SARS-CoV-2 lineages in wastewater. Temporal patterns in wastewater of lineage abundances in a mid-size municipality in Connecticut matched those defined by compiled clinical sequence data, and sequencing metrics were interrogated to define the Ct value and other confidence thresholds that ensure optimal performance. While this was evaluated specifically for a mid-size municipality, we do not see any reason why our approach would not work for larger-sized municipalities. We further show that this technique can be used with other wastewater sequencing techniques by expanding to samples taken across the USA in a similar timeframe. We find high variability in signal among individual samples, and limited ability to detect the presence of variants with clinical frequencies <10%; nevertheless, the overall trends match what we observed from sequencing clinical samples. In settings across the world where strong clinical sequencing programs do not exist, wastewater sequencing can be an effective tool for low-cost, efficient

monitoring of lineage abundance. As this is unlikely to be the last viral pandemic, nor the last with variants of concern, extending these approaches to other viruses and other sample types may allow broader monitoring of real-time pandemic evolution.

## Methods

### Constructing a reference set

We selected representative genomes per lineage from the GISAID database [5], downloaded on 9 March 2021. As our samples are from wastewater collection systems across the USA, we considered only reference sequences of US origin. After removing low-quality sequences (defined as having less than 29,500 non-ambiguous nucleotides) we randomly selected 1000 sequences per lineage for further analysis. We used minimap2 and paftools to align each of these sequences to the reference genome (MN908947.3) and subsequently identify variation with respect to this reference [31]. We then used VCFtools [32] to compute allele frequencies within each lineage. Based on these allele frequencies, we selected sequences per lineage such that all mutations with an allele frequency of at least 50% were captured at least once. This resulted in a final reference set of 1488 complete SARS-CoV-2 genome sequences.

### Designing benchmarking experiments

To evaluate the accuracy of our lineage abundance predictions, we created benchmarks consisting of a selection of non-variant genomes (background) and one variant lineage. In order to build benchmarks that reflect our real data as closely as possible, we selected the background genomes by taking all 11 sequences in GISAID collected in Connecticut on 2021-02-11 (the most recent collection date). Furthermore, we ensured that the sequences selected for generating benchmarks were not present in the reference set used for kallisto. For Spike-only benchmarks, we trimmed these sequences to keep only the Spike gene and simulated paired-end ($2\times150$ bp) Illumina sequencing reads at equal abundance using ART [33]. In addition, we randomly selected lineage sequences from GISAID and simulated sequencing reads for the Spike gene of each variant lineage (B.1.1.7, B.1.351, P.1, B.1.427, B.1.429) at varying frequencies (0.05, 0.06 ..., 0.1, 0.2, ..., 1, 2, ..., 10, 20, ..., 100%) to create 33 data sets per variant, hence 165 data sets in total. We performed these simulations at a total coverage of $100\times$ and $1000\times$ for whole genome benchmarks, and at $100\times$, $1000\times$, and $10,000\times$ for Spike-only benchmarks.

### Wastewater collection and sequencing from New Haven, CT

Primary sewage sludge samples were collected from the New Haven, CT, USA Wastewater Treatment Plant. The plan serves 200,000 residents in the towns of New Haven, Hamden, East Haven, and Woodbridge, CT. Primary sludge samples were collected from the effluent pump of the plant's gravity thickener. Samples were collected every other day starting January 3, 2021, and ending April 27, 2021. RNA was extracted from a 500-μL sample using a Zymo *Quick*-RNA Fecal/Soil Microbe Microprep Kit modified by the addition of 100 μL of phenol-chloroform to the bead beating step, and eluted in 50 μL of nuclease-free water. PCR cycle threshold (Ct) values of SARS-CoV-2 from undiluted sludge samples ranged from 30.2 to 35.3 ($7.1x10^3$ - $1.6x10^5$ virus copies / mL), indicating that there was enough viral RNA to apply genomic sequencing. We generated 400nt

tiled amplicons encompassing the SARS-CoV-2 genome using the Illumina COVIDSeq Test (RUO), modified to use NEBNext ARTIC V3 SARS-Cov-2 Primer Mixes 1 and 2 (instead of the included primer mixes) to improve genome coverage at low RNA concentrations [34, 35]. This included cDNA synthesis, amplicon generation, tagmentation, and cleaning. The pooled and cleaned library was sequenced on an Illumina NovaSeq at the Yale Center for Genomic Analysis; each sample was given at least 1 million reads of length $2\times150$ bp. Negative controls were included at the cDNA synthesis and amplicon generation steps. Sequencing depth was highly variable across the genome, ranging between 0 and more than $100,000\times$ (Fig. S8, Additional file 1).

Variant abundance of B.1.1.7 (Alpha) was quantified through a multiplex one-step RT-ddPCR assay developed by Bio-Rad Laboratories which targeted the H69/V70 deletion (Bio-Rad dMDS944624402). The H69/V70 deletion was unique to B.1.1.7 during the period of sampling. PCR conditions were as follows: 60 min at 50°, 10 min at 95°, 40 cycles of 30 s at 95°, and 1 min at 55°, followed by 10 min at 98° (Bio-Rad dMDS944624402). Automatic thresholding was used for the determination of positive droplets. Positive oligonucleotide controls were manufactured by Integrated DNA Technologies and included in all runs. No template negative controls were included in all runs. Abundance was calculated using the concentration of H69/V70 deletion detected (FAM labeled) divided by the sum of H69/V70 deletion and non-deletion type (HEX labeled). The confidence interval for the abundance was calculated from the ddPCR confidence intervals for the H69/V70 deletion and wild type concentrations as follows: lower_bound_abundance = lower_bound_ddPCR_del/(lower_bound_ddPCR_del + upper_bound_ddPCR_wildtype) and upper_bound_abundance = upper_bound_ddPCR_del/(upper_bound_ddPCR_del + lower_bound_ddPCR_wildtype).

### Wastewater collection and sequencing from across the US

Composite influent samples were collected by participating wastewater treatment facilities using equipment that these facilities already had in-house. Composite samples were aliquoted into three 50-mL conical tubes and shipped within 24 hours of collection overnight with ice packs to the Biobot Analytics laboratory (Cambridge, MA). Received samples were immediately pasteurized at 60°C for 1h.

One of the three tubes was then filtered to remove large particulate matter using a 0.2-µM vacuum-driven filter (EMD-Millipore SCGP00525 or Corning 430320, depending on sample turbidity). We then used Amicon Ultra-15 centrifugal ultrafiltration units (Millipore UFC903096) to concentrate 15mL of wastewater approximately $100\times$. We lysed viral particles in the concentrate by adding AVL Buffer containing carrier RNA (Qiagen 19073) to the Amicon unit before transfer and >10-min incubation in a 96-well 2mL block. To adjust binding conditions, 100% ethanol was added to the lysate, and samples were applied to RNeasy Mini columns or RNeasy 96 cassettes (Qiagen 74106 or 74181). For a subset of samples (all from locations within Massachusetts) we processed 45mL of wastewater by loading the same Amicon Ultra-15 unit three times.

The RNA samples resulting from the extraction process described above were used as the template for reverse-transcription (RT) reactions performed with LunaScript RT SuperMix enzyme mix (NEB) to generate cDNA. Reaction conditions were as follows: primer annealing at 25°C for 2 min, cDNA synthesis at 55°C for 10 min and heat

inactivation at 95°C for 1 min. Multiplexed polymerase chain reaction (PCR) amplification of cDNA was performed with Q5 Hot Start High-Fidelity 2X Master Mix (NEB) and ARTIC v3 primers (0.015 μM each, final) in two non-overlapping pools with the following cycling conditions: heat activation at 98°C for 30 s, followed by 35 cycles of 15 s denaturation at 98°C, 5 min annealing/elongation at 65°C.

The non-overlapping amplicon pools were combined and sequencing libraries for Illumina platform were prepared using tagmentation with bead-linked transposomes (Illumina) and a modified amplification protocol with KAPA HiFi HotStart ReadyMix (Roche) and combinatorial dual-indexed adapter sequences. Libraries were sequenced with NextSeq550 (Illumina) generating paired-end reads of 2×75 bp.

### Wastewater data preprocessing

Before processing with kallisto in our VLQ pipeline, we first removed adapter sequences from the reads using Trimmomatic [36], aligned the trimmed reads to a reference genome (GenBank MN908947.3) with BWA-MEM v0.7.17 [37], and subsequently identified primer sequences using iVar v.1.3.1 [18] and removed these with jvarkit (http://lindenb.github.io/jvarkit/Biostar84452).

### Clinical sequencing and data processing from New Haven, CT

#### *Ethics statement*

The Institutional Review Board from the Yale University Human Research Protection Program determined that the RT-qPCR testing and sequencing of de-identified remnant COVID-19 clinical samples obtained from clinical partners conducted in this study is not research involving human subjects (IRB Protocol ID: 2000028599).

#### *Sequencing and consensus generation*

Residual routine testing samples from confirmed SARS-CoV-2 positive individuals were provided by Yale New Haven Hospital, Yale Pathology Laboratory, "Yale Campus Study," Connecticut Department of Public Health, and Murphy Medical Associates. Sample types included nasal swabs in viral transport media, raw saliva, and extracted RNA. Samples not arriving as RNA were processed using the MagMAX viral/pathogen nucleic acid isolation kit; RNA was extracted from 300 μL of sample and eluted in 75 μL elution buffer. All products were tested using a locally developed assay for variants to determine viral RNA concentration [38]. Samples with sufficient RNA for sequencing (defined as a viral target cycle threshold value <35) were prepared using the Illumina COVIDSeq Test RUO for cDNA synthesis, amplicon generation, tagmentation, and cleaning. Pooled and cleaned libraries were sequenced using a 2×100 or 2×150 approach on an Illumina NovaSeq at the Yale Center for Genomic Analysis; each sample was given at least 1 million reads. Negative controls were included at RNA extraction, cDNA synthesis, and amplicon generation steps.

Reads were aligned to a reference genome (GenBank MN908937.3) using BWA-MEM v.0.7.15 [37]. Adaptor trimming, primer sequence masking, and simple majority base calling were conducted using iVar v1.2.1 [18] and SAMtools [39]. Lineages were assigned using pangolin v.2.4.2 [40].

## Supplementary Information

---

**Additional file 1.** Includes all supplementary information, supplementary figures and supplementary tables.

**Additional file 2.** Review history.

---

### Acknowledgements

James McGann and Jim Griffin were involved in developing the sequencing methodology at Ginkgo Bioworks.
Yale SARS-CoV-2 Genomic Surveillance Initiative authors
Ahmad Altajar, Anderson F. Brito, Anne E. Watkins, Anthony Muyombwe, Caleb Neal, Chen Liu, Christopher Castaldi, Claire Pearson, David R. Peaper, Eva Laszlo, Irina R. Tikhonova, Jafar Razeq, Jessica E. Rothman, Jianhui Wang, Kaya Bilguvar, Linda Niccolai, Madeline S. Wilson, Margaret L. Anderson, Marie L. Landry, Mark D. Adams, Pei Hui, Randy Downing, Rebecca Earnest, Shrikant Mane, Steven Murphy

### Peer review information

### Review history

The review history is available as Additional file 2.

### Authors' contributions

J.A.B., W.P.H., N.D.G., J.P., and M.B. conceptualized the project. A.Z., I.M.O., M.E.P., T.A., J.R.F., C.C.K., C.B.F.V., M.I.B., and the teams at Biobot and Ginkgo performed wastewater analysis and sequencing experiments. J.A.B., I.N., and M.J.L. executed the computational analyses. J.A.B., A.Z., I.M.O., M.E.P., C.D., W.P.H., N.D.G., J.P., and M.B. wrote the manuscript. All authors read, edited, and confirmed the final manuscript.

### Funding

This work was supported in part by the Pew Charitable Trusts, the David and Lucile Packard Foundation, NIH NIGMS award R35GM133700, and the Alfred P. Sloan Foundation (J.A.B. and M.B); CTSA Grant Number TL1 TR001864 (M.E.P. and T.A.); Fast Grant from Emergent Ventures at the Mercatus Center at George Mason University (N.D.G.); CDC Contract #75D30120C09570 (N.D.G.); Yale CoReCT pilot award (J.P. and N.D.G.); and NIH NIGMS award U54GM088558 (W.P.H.).

### Availability of data and materials

The raw SARS-CoV-2 sequencing data from New Haven wastewater (.fastq files) are available on NCBI SRA under Bioproject PRJNA741211 [41]. The clinical sequencing data can be accessed via covidtrackerct.com. The raw SARS-CoV-2 sequencing data from across the U.S. (.fastq files) are available on NCBI SRA under Bioproject PRJNA759260 [42]. The simulated wastewater sequencing data (.fastq files) for benchmarking are available on Zenodo (https://doi.org/10.5281/zenodo.5307070) [43]. All code used for the analysis presented in this manuscript is publicly available (MIT license) at https://github.com/baymlab/wastewater_analysis [44] and on Zenodo (https://doi.org/10.5281/zenodo.6612420) [45].

## Declarations

### Ethics approval and consent to participate
Not applicable.

### Consent for publication
Not applicable.

### Competing interests
N.D.G. is an infectious diseases consultant for Tempus Labs. W.P.H. is a scientific advisory board member to Biobot Analytics and has received compensation for expert witness testimony on the expected course of the pandemic. N.G. is a co-founder of Biobot Analytics; C.D., K.A.M., and M.I. are employees of Biobot Analytics.

### Author details

[1]Department of Biomedical Informatics, Harvard Medical School, Boston, MA, USA. [2]Department of Intelligent Systems, Delft University of Technology, Delft, Netherlands. [3]Department of Chemical and Environmental Engineering, Yale University, New Haven, CT, USA. [4]Department of Epidemiology of Microbial Diseases, Yale School of Public Health, New Haven, CT, USA. [5]Department of Epidemiology, University of Nebraska Medical Center, Omaha, NE, USA. [6]Biobot Analytics, Inc., Cambridge, MA, USA. [7]Ginkgo Bioworks, Inc., Boston, MA, USA. [8]Center for Communicable Disease Dynamics and Department of Epidemiology, Harvard T.H. Chan School of Public Health, Boston, MA, USA. [9]Department of Ecology and Evolutionary Biology, Yale University, New Haven, CT, USA.

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

## 

