## [**Additional file 2.** Review history. · Genome Biology]

Review History

First round of review

Reviewer 1

Were you able to assess all statistics in the manuscript, including the appropriateness of statistical tests used? There are no statistics in the manuscript.

Were you able to directly test the methods? No.

Comments to author:

1 -- Title is confusing due to the fact that RNA-Seq is mentioned. I would recommend using some alternative. For example transcriptome quantification

2 -- Abstract and main text. Variant should be define in the abstract. Does it mean a SNP or entire genome (genomic variant)

3 -- Abstract. "This technique". Which technique?

4 -- other wastewater sequencing. More details are need. What is other? Other technology, protocol. It is still unclear what is original sequencing and other are?

5 -- Please mention some metric in the abstract like PPV/sensitivity or accuracy

6 -- Introduction. Line 45. "Our approach". Define your approach

7 -- How proposed bioinformatics approach differs from ref 11,12

8 -- Introduction line 71. "Specific lineages". Which lineages. More details are needed

9 -- Authors mention the analogy of the problem with transcriptome quantification problem. Which is correct. However it is more similar to viral quasispecies quantification problem. This needs to be acknowledge and corresponding literature cited

For example

<https://www.ncbi.nlm.nih.gov/pmc/articles/PMC5467126/>

<https://academic.oup.com/bioinformatics/article/30/12/i329/392298?login=true>

10 Result section line 88. "Different infections". It is confusing. Do the authors mean viral genomes?

11 more insight into performance of Kalisto will be helpful. For example number of reads utilized by Kalisto. Also limitation of pseudo alignment needs to be acknowledge compared to alignment based methods. For example one limitation is that only uniquely mapped reads are assigned

It can be helpful to cite a recent review on this topic

<https://pubmed.ncbi.nlm.nih.gov/34446078/>

12 It will be helpful to discuss how using a other type of quantification methods can help

For example RSEM and ISOEM

<https://bmcbioinformatics.biomedcentral.com/articles/10.1186/1471-2105-12-323>

<https://almob.biomedcentral.com/articles/10.1186/1748-7188-6-9>

13 -- How % for filtering was determined. It would be nice to see the effect of that on Accuracy/PPV/Sensitivity. For example a plot metric vs all variants with frequency above x%

14 -- It would be interesting to know how the proposed approach compares to other algorithms. For example a baseline where ne maps reads to viral genomes and considers only uniquely mapped reads also it would be interesting to know to this compares to EM-based algorithms RSEM and IsoEM

15 -- Authors mention reads in wastewater water is typically poor and with large variability in coverage. Is this accounted for in simulations?

One approach can be to learn the coverage trends in real data and incorporate this in simulated data. Additionally did the authors vary the read length and error rate in the data?

In general, this is important issues and needs to be discussed in Discussion section.

16 -- Is it possible to generate the real benchmark similar mock communities are produced? It would be interesting to discuss this this in Discussion Section

17 -- Results presented in line 160 - 184 are extremely hard to follow. Authors may consider to re write it to improve the clarity

18 - Additional experiments are required to further understand the effect of close related strains on ability of kalisto or other methods to accurately assigned the reads to correct strain. One approach is to vary the Edit distance between strain in teh sample and see the effect of that on the performance

Similar was done previously here for viral data

<https://genomebiology.biomedcentral.com/articles/10.1186/s13059-020-01988-3>

19 -- What are the general trends in estimation. Is proposed method tends to under or over estimate the relative abundances?

20 -- Lines 199 -- 204 belong to discussion

21 Lines 255 - 237 This is too technical for results section. Some of details belongs to methods section.

22 -- It is hard to understand what was real data. How many samples were sequenced in house vs public data. Also how the read length and sequencing depth varied between in house data and public data

23 -- line 234 Authors mention number of reads and genome coverage. Why not incorporate this in simulated benchmark

24 Line 290. Section "Wastewater abundance ..." Does this section talk about in house or in house and public data? It is hard to understand

25 -- Line 334. How 10% was determined?

26 -- It would be nice to explain in Introduction or discussion what was the purpose to generate Spike only sequencing and how this data is different from WGS?

27 Line 376. Authors talk about mid size municipalities. It would be interesting to hear authors opinion on suitability of proposed approach for large sie municipality. Some previous work is here

<https://pubs.rsc.org/en/content/articlelanding/2021/ew/d1ew00586c>

28 Authors provide code used for analysis in github repo. This is fantastic! To enable full reproducibility of produced results, it would be also nice to provide the code and data behind the figures (both main ones and suppl). See recommendations here

<https://osf.io/uz7m5>

29 - It would be helpful to show data points in Figure 3c behind the boxplot

Minor comments:

1 -- it would be helpful to cite a recent review about genomics opportunities of genomics for pandemics response

For example

<https://www.ncbi.nlm.nih.gov/pmc/articles/PMC8109901/>

2- define lineage in the introduction and explain how it is different from strain and genomic variant

3 Figure 2 Instead of top, it would be easier to add labels - a b c

In the figure and reference the figure

As Figure 2a 2b etc

4 -- Figure 2. Please define VOC in caption of figure. PLease add labels to each panel (a bc d) And describe each panel in caption

5 -- Line 425 "across US". It would be interesting to know which cities? Also was the sample obtained from facilities serving different number of people

Reviewer 2

Were you able to assess all statistics in the manuscript, including the appropriateness of statistical tests used? Yes, and I have assessed the statistics in my report.

Were you able to directly test the methods? No.

Comments to author:

The article describes a repurposing of an existing software, Kallisto, for a reference-based estimating abundances for a set of predefined SARS-COV-2 viral variants of concern from simulated benchmark data and wastewater data. An analysis of variant abundances estimation is performed on simulated benchmark data and "clinical abundances" estimated from GISAID data

and clinical samples. It is a well written article with nice illustrations. However, the article demonstrates the value of this workflow for the intended application not in a convincing manner, as (i) the benchmark data are rather simple and unrealistic, (ii) a comparison with other methods is lacking and (iii) there are inherent limitations to the approach, which - contrary to other approaches - is unable to detect and quantify novel variants not included in the reference collection.

1) There is no methodological innovation, except for a repurposing of the Kallisto software. Originally, kallisto was designed for quantifying abundances from RNA-seq data and is used here on viral RNA from wastewater sequencing.

2) No new variants or - if I understood this right - even existing variants not included in the manually selected Kallisto reference collection - can be detected with this approach. This seems like a severe limitation certainly for an early monitoring system using wastewater. For instance, if omicron were present in those wastewater samples, it would not be detectable. Other methods exist, also de novo inference ones, that do not have this limitation. For instance, there are de novo assembly-based abundance estimation techniques, with methods like SAVAGE (by the same first author) or Haploflow, which assemble the data first and give an estimation of the abundances. The authors do not show a comparison to these tools, though they have been demonstrated to be applicable to this problem - is there a reason for this?

3) While the authors mention other reference-based methods like V-pipe and iVar, there is no comparison to the performance of these methods. The authors claim "[...] identifying individual mutation frequencies from wastewater sequencing data is highly error-prone---any errors would be propagated into variant abundance estimates" but do not provide evidence for this. A comparison of the pipeline against these popular tools, particularly on the simulated data, seems warranted.

4) The benchmarking could be extended to include assessment on simulated test data that is more realistic and differs in composition more substantially from the Kallisto training data: lines 198-204: "However, our benchmarks are generated using a single variant sequence, while real data consists of a mixture of different sequences for the same variant." I agree that these data sets are likely not very realistic for performances on real data consisting of mixtures of different variants.

5) How were sequencing errors included in the benchmark? What ART model was used to simulate sequencing errors?

6) How would the workflow perform on data sets including variants not part of the manually selected reference set, e.g Omicon?

7) How would the workflow perform for known variants carrying additional changes not included in the reference collection?

The authors write that there is overestimation of low abundant and underestimation for higher abundant variants (Lines 264-269). While this can be expected, I would have wished for a

reference-based method to have less problems of this sort. The authors write that "it would be interesting to evaluate predictions on real data more thoroughly, e.g., by comparing to qPCR-based variant abundance estimates per variant." This benchmark should have been performed.

8) It is nice to see that the abundances of variants in wastewater corresponds to the "real" GISAID abundances, but it does not come as much of a surprise either and has been shown before, e.g. in 10.3201/eid2705.204410. There is also an issue considering that GISAID data, if not coming from systematic surveillance efforts and sequencing of random samples, is not representative for population level viral variant abundances, which should be discussed.

Additional comments:

It is not entirely clear what FPR and FNR refers to in Table 1. It seems to be the prediction of non-existent strains or the non-prediction of existent strains, but it is not explained anywhere. There seems to be a correlation between Ct value and genome coverage, but there is no correlation coefficient (lines 234-236). It is of no surprise that there are less genomes with >20x coverage if the number of aligned reads is low (Figure 3b).

Since some variants are present with up to 17 sequences and others with only a single sequence, it would make sense to include a test data set which shows that there is no bias towards the variants overrepresented in the reference database.

The comparison shown in Figure 5 is interesting, but stating that the variant abundances match the expected pattern seems not entirely evident, even though the authors state that "individual samples are unreliable": Particularly for B.1.429, there is no overlap between Wastewater and GISAID and even the trend for the others does not match really well.

In addition to what is there the Github repository should include a requirements.txt or something similar for ease of installation. Right now every single software has to be installed manually via trial and error. After this is done, the example runs and produces the expected result.

Additionally, problems specific for viral abundance estimation from RNA-seq data as e.g. described here <https://journals.asm.org/doi/pdf/10.1128/JVI.01342-18> are not mentioned.

Response to reviewer #1

Thank you for carefully reading our manuscript and providing detailed feedback. We have addressed each of the comments and provide further explanations below.

Title

1 -- Title is confusing due to the fact that RNA-Seq is mentioned. I would recommend using some alternative. For example transcriptome quantification

Thanks for pointing this out, we have changed the title to avoid further confusion.

Abstract

2 -- Abstract and main text. Variant should be defined in the abstract. Does it mean a SNP or entire genome (genomic variant)

Excellent point, while the term “variant” is commonly used to refer to SARS-CoV-2 mutants, it turns out to be a confusing term because of the different meaning in general bioinformatics, where it refers to a single mutation. We have therefore decided to avoid the term “variant” as much as possible throughout the manuscript and we do not use it in the abstract anymore.

3 -- Abstract. "This technique". Which technique?

We mean our computational pipeline, we have now clarified this.

4 -- other wastewater sequencing. More details are needed. What is other? Other technology, protocol. It is still unclear what is original sequencing and other are?

We had to drastically shorten the abstract to adhere to the maximum of 100 words, so this sentence has been removed altogether.

5 -- Please mention some metric in the abstract like PPV/sensitivity or accuracy

Given the limited word count allowed for the abstract, unfortunately this leaves no room for detailed results; we do discuss such metrics in the results section and discussion.

Introduction

6 -- Introduction. Line 45. "Our approach". Define your approach

By “our approach” we meant the overall approach to pandemic management, i.e. appropriate measures and public health strategies to be implemented. We understand the confusion, hence we have rephrased this sentence:

"For optimal pandemic management and public health strategies, we need first to be able to observe which lineages are present where, and critically how the relative abundance of these lineages is changing in the population."

7 -- How proposed bioinformatics approach differs from ref 11,12

References 11 and 12 make use of available pipelines for SNP calling to identify individual mutations across the viral genome—they do not provide methods to predict lineage abundance. The point that we are trying to make after referencing 11 and 12, is that based on the fact that these papers have shown that quantifying relative abundances of specific mutations in wastewater samples is possible, it may also be possible to impute relative abundance of viral lineages (defined by a set of characteristic mutations). Our approach is one of the first steps towards making this possible. We have adapted the text to clarify this:

"The genomes of SARS-CoV-2 in wastewater can also be sequenced, which can then be used to identify mutations present in an entire community with respect to the reference genome^{12,13}."

"Given the urgency of wastewater surveillance during the ongoing pandemic, several tools are under development to tackle the problem of lineage (or variant) quantification from wastewater sequencing data. For example, Ellmen et al. define an optimization problem to combine individual mutation frequencies into lineage frequencies¹⁵ and Karthikeyan et al. propose an algorithm based on a regression problem to minimize the edit distance between sequences and a reference¹⁶. Each of these approaches under development relies on the discovery and quantification of individual mutations using popular tools like V-pipe¹⁷ or iVar¹⁸, a process that is highly error prone given the nature of wastewater sequencing data."

8 -- Introduction line 71. "Specific lineages". Which lineages. More details are needed

By specific we meant to say any lineage of interest, as specified by the user. We have rephrased this sentence:

"(..) and suggests the possibility of monitoring the abundance of lineages of concern or interest".

9 -- Authors mention the analogy of the problem with transcriptome quantification problem. Which is correct. However it is more similar to viral quasispecies quantification problem. This needs to be acknowledged and corresponding literature cited. For example

<https://www.ncbi.nlm.nih.gov/pmc/articles/PMC5467126/>

<https://academic.oup.com/bioinformatics/article/30/12/i329/392298?login=true>

This is an excellent point, which we now explain further in our manuscript (lines 110–118). Indeed, the problem is very similar to the viral quasispecies quantification problem. However, an important distinction here is that the viral quasispecies quantification problem typically refers to clinical samples, which are of much higher quality than wastewater samples: wastewater samples suffer from RNA degradation, inhibitory compounds, low RNA concentrations and strong amplicon bias, etc. As a consequence, regular viral quasispecies assembly/quantification tools are not suitable for this type of data. We have added the following paragraph to the manuscript:

"It is important to note that the computational task of lineage abundance prediction from wastewater sequencing data is also highly similar to the viral quasispecies quantification problem. There, the input consists of sequencing data from a single infection where within-host evolution has resulted in a so-called quasispecies: a cloud of closely related mutants that together comprise the infection¹⁹. While this input seems similar to wastewater sequencing data, the crucial difference is that wastewater samples suffer from factors such as inhibitory compounds, RNA degradations, low RNA concentrations and frequent amplicon drop-out. Per consequence, regular viral quasispecies quantification approaches (see Eliseev et al.²⁰ for an overview) are not suitable for this type of data---instead, we need an approach that is robust to the variability in wastewater sequencing data quality."

Results

10 Result section line 88. "Different infections". It is confusing. Do the authors mean viral genomes?

Thank you for pointing this out, we understand the confusion. What we meant to say is that the SARS-CoV-2 genome fragments found in wastewater do not come from a single person that is infected, but from an entire subset of the population (all infected persons within a certain geographical region). We have clarified this in the text (lines 95–96) to avoid further confusion:

"SARS-CoV-2 RNA fragments in wastewater originate from different people carrying a SARS-CoV-2 infection, with potentially different viral lineages, pooled together into a single sample."

11 more insight into performance of Kallisto will be helpful. For example number of reads utilized by Kalisto. Also limitation of pseudo alignment needs to be acknowledge compared to alignment based methods. For example one limitation is that only uniquely mapped reads are assigned. It can be helpful to cite a recent review on this topic

<https://pubmed.ncbi.nlm.nih.gov/34446078/>

We now provide additional alignment statistics for kallisto in the supplementary material (Section 5.3). We also discuss the difference between pseudo-alignment and regular alignment in the paragraph where we discuss other transcript quantification approaches (lines 256–266). However, we believe that pseudo-alignment is not a limitation but rather a strength of our approach: during pseudo-alignment with kallisto, reads are not assigned exact locations, only compatibility with available transcripts is evaluated. Hence, a read does not need to align to a transcript uniquely. In regular read alignment, many reads can align to many transcripts, and the read aligners cannot handle this: the alignment process becomes very slow and the alignment files become extremely large. Instead of keeping unnecessarily large SAM/BAM files, kallisto only keeps compatibility classes. This is beneficial to runtime, memory usage and storage space needed, and as a result, kallisto does not need to limit the number of hits allowed per read (contrary to read-to-transcriptome aligners). We have added a discussion on this topic in the supplementary material (Section 5.2).

12 It will be helpful to discuss how using other types of quantification methods can help. For example RSEM and ISOEM.

<https://bmcbioinformatics.biomedcentral.com/articles/10.1186/1471-2105-12-323>
<https://almob.biomedcentral.com/articles/10.1186/1748-7188-6-9>

We have added a paragraph on other transcript quantification tools to the results section (lines 255–266):

"While for this study we primarily used kallisto²¹, we also evaluated performance for other transcript quantification tools. First, we evaluated the software package salmon²², which takes a slightly different algorithmic approach to the same problem (Fig S10, Additional File 1) and found predictions were highly similar to those obtained with kallisto, the main difference being that salmon is slightly more conservative: it achieves higher precision (fewer false positives), at the expense of lower recall (more false negatives). Second, we evaluated RSEM²⁴ and Iso-EM2²⁵, both of which are EM-based algorithms based on regular read-to-transcriptome alignment²⁶. While RSEM consistently overestimates lineage abundance, Iso-EM2 consistently underestimates abundance (Table S2, Additional File 1). Overall, kallisto performs best among this selection of popular transcript quantification algorithms. Although salmon tends to miss lineages at very low frequencies, one potential advantage is that this method may also be applied to long reads (in alignment-based mode), while kallisto usage is limited to short reads."

13 -- How % for filtering was determined. It would be nice to see the effect of that on Accuracy/PPV/Sensitivity. For example a plot metric vs all variants with frequency above x%

The filtering threshold was determined experimentally: across datasets, we typically observe a noise level up to $\sim 0.1\%$ (see also supplementary figure S2). Clearly, increasing this threshold would further denoise the predictions, but we may also lose valuable information. In particular, we would lose the ability to predict lineages of very low abundance. This is why we decided on a conservative threshold of 0.1%, but in practice anything between 0.1 and 1% would be fine. As we show in Table 1, increasing the threshold to 1% leads to higher precision at the cost of slightly lower recall compared to a threshold of 0.1%. In the end, the optimal choice depends on the application: if the purpose is to find lineages of low abundance, it is safer to use a conservative threshold of 0.1%, while a focus on predicting lineages at higher abundance would warrant a threshold of 1%. This is up to the user, the filtering threshold value can be specified when running our pipeline. We have added a discussion of this point in the text (lines 162–174 and 248–249).

"On our simulated datasets, where we know the true frequency of each lineage, we observe a background noise of 0.01--0.09% (Fig S2, Additional File 1), meaning that some sequences are falsely predicted to be present at 0.01--0.09% abundance. These false positives are likely due to shared mutations or conserved sequences between lineages. The level of background noise tends to be higher for whole genome benchmarks than for Spike-only benchmarks, because the majority of defining lineage mutations are in the Spike gene (Fig S1, Additional File 1 and Fig 2). In both cases, we apply a threshold of 0.1% abundance to include a sequence in the lineage abundance computation and we only report the presence of a lineage exceeding this threshold to avoid false positives."

"Increasing the minimal abundance thresholds reduces the false positive rate but increases the false negative rate."

14 -- It would be interesting to know how the proposed approach compares to other algorithms. For example a baseline where we map reads to viral genomes and considers only uniquely mapped reads also it would be interesting to know how this compares to EM-based algorithms RSEM and IsoEM

We have now performed additional experiments to compare kallisto to RSEM and IsoEM2, results can be found in supplementary table S2. In general, IsoEM2 severely underestimates while RSEM severely overestimates lineage abundance. We briefly discuss this comparison in the manuscript (lines 261–263) and we provide a more detailed analysis in the supplementary material (Section 5.2).

Discussion

15 -- Authors mention reads in wastewater water is typically poor and with large variability in coverage. Is this accounted for in simulations? One approach can be to learn the coverage trends in real data and incorporate this in simulated data. Additionally did the authors vary the read length and error rate in the data? In general, this is important issues and needs to be discussed in Discussion section.

These are excellent suggestions. While the benchmarking data presented in the manuscript already accounts for some variability in coverage, it does not reflect the extreme variability (amplicon dropout) observed in real wastewater sequencing data. Hence, we have created additional benchmarking datasets and performed additional experiments to evaluate the impact on our lineage abundance predictions. For full details and results please see the supplementary material, section 3.2. These experiments show that amplicon dropout and extreme variability in coverage has relatively little impact on the accuracy of our predictions.

In addition, we have looked into the impact of sequencing error rates. In this analysis we distinguish between substitution errors, insertion errors, and deletion errors. It turns out that insertion and deletion errors have very little impact on prediction accuracy. Substitution errors, however, do impact predictions. The explanation for this is simple: due to high diversity between lineages, a substitution error will often match another lineage, thus leading to misclassification of the corresponding read and hence underestimation of the corresponding lineage. However, we also observe that at regular sequencing error rates (<1%) the impact of substitution errors on prediction accuracy remains very limited. Please see the supplementary material, section 4, for a detailed description and results for these experiments.

Finally, the suggestion of read length is very interesting: clearly, longer reads would be beneficial to the accuracy of lineage abundance predictions, since reads can span larger regions and hence align (more) uniquely to the reference genomes. However, it is likely less applicable to wastewater sequencing: the fragments that we sequence are so short, that longer read lengths are unrealistic under our sequencing protocol. Hence at this point we consider it beyond the scope of this manuscript to investigate the impact of read length on prediction accuracy; instead, we chose to simulate reads that resemble our real sequencing data as closely as possible, also in terms of read length.

We summarize and discuss these findings in the Discussion:

"As part of our benchmarking experiments, we have attempted to incorporate the variability in sequencing depth (amplicon bias) in our simulated data and we have investigated the impact of sequencing errors on prediction accuracy. Both factors were shown to have little impact on the accuracy of our VLQ pipeline, provided that the sequencing error rate is within the expected range for Illumina sequencing (0.1--1%). It would be interesting to explore the use of long-read sequencing for wastewater analysis; in practice, however, RNA fragments may be too short to allow for high-throughput long-read sequencing due to RNA degradation in wastewater. For future work, it would be valuable to generate real benchmarking data by creating artificial mixtures of lineages at pre-determined proportions, and evaluate robustness of the sequencing and computational steps."

16 -- Is it possible to generate the real benchmark similar mock communities are produced? It would be interesting to discuss this this in Discussion Section.

This is certainly an interesting suggestion, though we feel those experiments would be a research project of its own, and therefore beyond the scope of the paper. We however do now discuss this in the Discussion:

"For future work, it would be valuable to generate real benchmarking data by creating artificial mixtures of lineages at pre-determined proportions, and evaluate robustness of the sequencing and computational steps."

17 -- Results presented in line 160 - 184 are extremely hard to follow. Authors may consider to re write it to improve the clarity

We have rewritten these paragraphs, now providing additional guidance and insights for the reader (lines 176–213).

18 - Additional experiments are required to further understand the effect of closely related strains on the ability of kalisto or other methods to accurately assign the reads to correct strain. One approach is to vary the Edit distance between strain in the sample and see the effect of that on the performance.

Similar was done previously here for viral data

<https://genomebiology.biomedcentral.com/articles/10.1186/s13059-020-01988-3>

Thank you for the suggestion, this is indeed an interesting experiment. We performed it on a mixture of two lineages with increasing edit distance. Our main conclusion from this experiment is that the edit distance itself has little impact on the accuracy of our abundance predictions. Full details can be found in the supplementary material, section 3.1.

19 -- What are the general trends in estimation. Is proposed method tends to under or over estimate the relative abundances?

We discuss over- and underestimation issues in the paragraphs describing Figure 2. Based on comment 17 we have rewritten these parts to improve clarity (lines 176–213). To summarize, we observe consistent underestimation of lineage abundance predictions, which can potentially be corrected for. At low frequencies (<1%) we observe overestimation due to highly similar sequences being present in the data. Sequencing errors also contribute to this effect: without sequencing errors, we get near-perfect predictions (Supplementary figure S9). Please find a detailed discussion regarding the impact of sequencing errors in the supplementary material (Section 4).

20 -- Lines 199 -- 204 belong to discussion

We have now adapted this paragraph, as the qPCR experiments suggested were actually performed (see also reviewer 2, comment 7). While the original lines indeed were more appropriate for the discussion, we believe that this paragraph in its current shape (lines 331–340) belongs to the results section:

"To complement the clinical data of lineage frequencies, we have analysed our wastewater samples using ddPCR to estimate the prevalence of the B.1.1.7 lineage based on the presence of the characteristic H69/V70 deletion in the Spike gene (Fig 4). We observe that ddPCR-based abundance estimates are highly variable. While at some time points the ddPCR-based abundance estimates are very close to the predictions obtained with kallisto, at other time points these estimates are quite different. We investigated these specific instances by aligning the sequencing reads to the SARS-CoV-2 reference genome (MN908947.3) and analyzing the alignments across the Spike deletion at amino acid positions 69-70 (as targeted by ddPCR). We observed that in these sequencing data sets, this specific region is not even covered by sequencing reads, thus confirming our earlier observation that wastewater sequencing data is highly stochastic."

21 Lines 225 - 237 This is too technical for results section. Some of details belongs to methods section.

Thank you for pointing this out, we have moved the technical details to the methods section (lines 510–517).

22 -- It is hard to understand what was real data. How many samples were sequenced in house vs public data. Also how the read length and sequencing depth varied between in house data and public data.

All sequencing data discussed in the manuscript is in-house data. The only public data used is the GISAID database to obtain reference sequences and to estimate lineage abundances in the population. We now state this explicitly in the manuscript (lines 357 and 535). Read length and sequencing depth are discussed in the methods section:

"The pooled and cleaned library was sequenced on an Illumina NovaSeq at the Yale Center for Genomic Analysis; each sample was given at least 1 million reads of length 2x150 bp. Negative controls were included at the cDNA synthesis and amplicon generation steps. Sequencing depth

was highly variable across the genome, ranging between 0 and more than 100,000x (Fig S11, Additional File 1)."

23 -- line 234 Authors mention number of reads and genome coverage. Why not incorporate this in simulated benchmark.

The simulated benchmarks were designed to reflect the real data as best as possible in all aspects, including sequencing depth. While we may not have explicitly stated this in the previous version, we now do (lines 148–160 and 489–499). However, it is important to note that there is a lot of variation in terms of sequencing depth between and within samples. Hence, we also created benchmarks where genome coverage (and amplicon dropout) represents real data (Supplementary material, section 3.2).

24 Line 290. Section "Wastewater abundance ..." Does this section talk about in house or in house and public data? It is hard to understand.

Same answer here as for comment 22: all sequencing data discussed in the manuscript is in-house data. We clarified this in the text (lines 357 and 535).

25 -- Line 334. How 10% was determined?

This is based on the results shown in Figure 4, where we see that for clinical frequencies below 10% the wastewater predictions are often lacking, while above 10% we see a clear signal. This is now explained in the Results section:

"Figure 4 suggests that the detection threshold for these variant lineages is around 10%: for clinical frequencies below 10% the wastewater-based predictions tend to be underestimated or lacking, while above this threshold we see a clear signal, most strongly for B.1.526."

26 -- It would be nice to explain in Introduction or discussion what was the purpose to generate Spike only sequencing and how this data is different from WGS?

Thank you for pointing this out, we have added our motivation and further discussion on this topic in the Discussion (lines 433–441).

"Since sequencing a smaller genomic region allows for sequencing at higher depth at the same cost, we evaluated prediction accuracy on Spike-only benchmarking data. The Spike gene is a particularly variable region in the SARS-CoV-2 genome, hence it is highly informative for the purpose of lineage quantification. In fact, with perfect coverage (simulated data) Spike-only sequencing gives better predictions than whole genome sequencing. More broadly, as the lineage-discriminating mutations are restricted to a subset of the amplicons in tiled sequencing, a strategy focusing sequencing depth proportionally to the potentially informativity of the tiles would likely yield the more accurate predictions per sequence read. In practice, whole genome sequencing is often necessary to obtain enough information versus restricting to Spike."

27 Line 376. Authors talk about mid size municipalities. It would be interesting to hear authors opinion on suitability of proposed approach for large sie municipality. Some previous work is here

<https://pubs.rsc.org/en/content/articlelanding/2021/ew/d1ew00586c>

Interesting question; the point of saying mid-sized municipalities in the manuscript is simply that this is what we tested primarily. The US datasets (Fig 5) include results for municipalities of different sizes, but for these datasets we do not have clinical data to evaluate our

predictions. However, we don't see any reasons why this wouldn't work for larger sized municipalities. We now also state so in our conclusions:

"Temporal patterns in wastewater of lineage abundances in a mid-size municipality in Connecticut matched those defined by compiled clinical sequence data, and sequencing metrics were interrogated to define the Ct value and other confidence thresholds that ensure optimal performance. While this was evaluated specifically for a mid-size municipality, we do not see any reason why our approach would not work for larger sized municipalities."

28 Authors provide code used for analysis in github repo. This is fantastic! To enable full reproducibility of produced results, it would be also nice to provide the code and data behind the figures (both main ones and suppl). See recommendations here <https://osf.io/uz7m5>

All code to reproduce our results, from simulating benchmarking data to running the pipeline and obtaining the figures is available in the github repo.

29 - It would be helpful to show data points in Figure 3c behind the boxplot

Outliers are now represented as data points on the plot.

Minor comments:

1 -- it would be helpful to cite a recent review about genomics opportunities of genomics for pandemics response
For example <https://www.ncbi.nlm.nih.gov/pmc/articles/PMC8109901/>

We have added this citation in the introduction (line 259).

2- define lineage in the introduction and explain how it is different from strain and genomic variant

We updated the introduction to include these definitions. "Strain" can be a complex term and we no longer use it in the manuscript.

"A SARS-CoV-2 lineage is a group of closely related viruses with a common ancestor defined by key mutations, some of which are likely to be selective adaptations of the SARS-CoV-2 virus. Some lineages, or groups of lineages, with public health interest are then classified as variants."

3 Figure 2 Instead of top, it would be easier to add labels - a b c

In the figure and reference the figure as Figure 2a 2b etc.

4 -- Figure 2. Please define VOC in caption of figure. Please add labels to each panel (a bc d)

And describe each panel in caption

Excellent suggestion, we have added labels to the subfigures in Figure 2 and now refer to these. We describe each panel in the caption and also define VOC there.

5 -- Line 425 "across US". It would be interesting to know which cities? Also was the sample obtained from facilities serving different number of people

These are indeed interesting details, but unfortunately this is information that cannot be shared for privacy reasons—we can only provide state-level information.

Response to reviewer #2:

The article describes a repurposing of an existing software, Kallisto, for a reference-based estimating abundances for a set of predefined SARS-COV-2 viral variants of concern from simulated benchmark data and wastewater data. An analysis of variant abundances estimation is performed on simulated benchmark data and "clinical abundances" estimated from GISAID data and clinical samples. It is a well written article with nice illustrations. However, the article demonstrates the value of this workflow for the intended application not in a convincing manner, as (i) the benchmark data are rather simple and unrealistic, (ii) a comparison with other methods is lacking and (iii) there are inherent limitations to the approach, which - contrary to other approaches - is unable to detect and quantify novel variants not included in the reference collection.

Thank you for your constructive feedback on our manuscript, we will address each of the comments in detail below. Let us briefly respond to the overall comments here:

(i) Our goal was for the benchmarks to resemble the real data as closely as possible; we have added many additional results in the supplementary material, including more extensive and challenging benchmarking experiments.

(ii) Indeed, a comparison to other approaches would be valuable, but as this is a very recent problem all relevant approaches are still under development. We have tried several, but all gave either installation or runtime errors that we were unable to resolve. We now added a comparison to other transcriptome quantification tools, to give further insight into our choice for kallisto. Future studies will be able to delve deeper into the comparison between different, newly developed variant quantification algorithms.

(ii) While our approach has its limitations, it also has several strengths compared to other approaches. We cannot detect new lineages, but as soon as a lineage has been observed once, we can keep track of it. The main strength of our approach is that we do not rely on SNP calling algorithms and we avoid bias towards a single reference genome. Instead, we use a large reference set of sequences, representing all known variation in SARS-CoV-2 genomes. We can even make use of within-lineage variation to distinguish a lineage from other, highly similar lineages. And finally, we do not only predict abundance for variant lineages, but we can predict the entire sample composition.

1) There is no methodological innovation, except for a repurposing of the Kallisto software. Originally, kallisto was designed for quantifying abundances from RNA-seq data and is used here on viral RNA from wastewater sequencing.

Indeed, the step from RNA-seq data to viral RNA in wastewater seems small, but at the same time it is a different field and a different type of data. The wastewater sequencing data that we generated here was not via traditional RNA-seq, rather by amplicon-based virus sequencing. The resulting data is quite different from RNA-seq data in terms of quality and sequencing protocols, so we cannot assume that transcriptome quantification methods "just work" for the variant quantification problem. We believe that our experimental results are very valuable to anyone interested in viral surveillance from wastewater sequencing data. While we do not provide major methodological innovations, we provide an entire workflow that is easy to run. Moreover, the application itself is novel and of high societal value.

2) No new variants or - if I understood this right - even existing variants not included in the manually selected Kallisto reference collection - can be detected with this approach. This seems like a severe limitation certainly for an early monitoring system using wastewater. For instance, if omicron were present in those wastewater samples, it would not be detectable. Other methods exist, also de novo inference ones, that do not have this limitation. For instance, there are de novo assembly-based abundance estimation techniques, with methods like SAVAGE (by the same first author) or Haploflow, which assemble the data first and give an estimation of

the abundances. The authors do not show a comparison to these tools, though they have been demonstrated to be applicable to this problem - is there a reason for this?

We would like to clarify that the purpose of our pipeline is to monitor prevalence of viral lineages throughout a pandemic. Detecting a novel mutant is not what our pipeline is aimed at and this is indeed not something that our pipeline is currently capable of. But as soon as a novel lineage has been detected somewhere, we can update our reference set to include this sequence and quantify abundance for the corresponding lineage. In this sense, our approach is highly flexible.

Regarding de novo assembly-based techniques, we have added a short discussion on this at the beginning of the results section (lines 110–118), after explaining the analogy to RNA transcript abundance estimation. There, we highlight the computational similarity to the viral quasispecies quantification problem, but also point out that these methods are not suitable for wastewater sequencing data. The crucial difference is that wastewater samples suffer from various factors that result in low data quality, such as inhibitory compounds, RNA degradation, low RNA concentrations and frequent amplicon drop-out. Viral quasispecies assembly approaches require full genome coverage at relatively stable sequencing depth to be able to reconstruct the genome and predict abundance. We have clarified this in the manuscript to avoid further confusion:

"It is important to note that the computational task of lineage abundance prediction from wastewater sequencing data is also highly similar to the viral quasispecies quantification problem. There, the input consists of sequencing data from a single infection where within-host evolution has resulted in a so-called quasispecies: a cloud of closely related mutants that together comprise the infection¹⁹. While this input seems similar to wastewater sequencing data, the crucial difference is that wastewater samples suffer from factors such as inhibitory compounds, RNA degradations, low RNA concentrations and frequent amplicon drop-out. Per consequence, regular viral quasispecies quantification approaches (see Eliseev et al.²⁰ for an overview) are not suitable for this type of data---instead, we need an approach that is robust to the variability in wastewater sequencing data quality."

3) While the authors mention other reference-based methods like V-pipe and iVar, there is no comparison to the performance of these methods. The authors claim "[...] identifying individual mutation frequencies from wastewater sequencing data is highly error-prone---any errors would be propagated into variant abundance estimates" but do not provide evidence for this. A comparison of the pipeline against these popular tools, particularly on the simulated data, seems warranted.

We realize that this sentence was rather unclear; what we meant to say was that recent research on SARS-CoV-2 diversity in wastewater and lineage quantification has focused primarily on individual mutation frequencies. These can be analysed with V-pipe or iVar, but these tools do not quantify lineages, and so are not directly comparable.

New algorithms aiming to tackle this problem are under development and we attempted to run these algorithms on our data, but unfortunately all gave either installation or runtime errors that we were unable to resolve. We expect that in a year or so, when these tools have been published and developed further, benchmarking experiments can be performed to properly compare performance. We have added additional references and further information to the main manuscript (lines 325–331).

4) The benchmarking could be extended to include assessment on simulated test data that is more realistic and differs in composition more substantially from the Kallisto training data: lines 198-204: "However, our benchmarks are generated using a single variant sequence, while real data consists of a mixture of different sequences for the same variant." I agree that these data sets are likely not very realistic for performances on real data consisting of mixtures of different variants.

Regarding the composition of the simulated test data and the reference set used for predictions with kallisto, please note that these do not contain the same sequences. In other words, there is no overlap between testing and training data. We also added this clarification in the manuscript (lines 492–493). Nevertheless, creating more realistic benchmarking data is an excellent suggestion. While the benchmarking data presented in the manuscript already accounts for some variability in coverage, it does not reflect the extreme variability (amplicon dropout) observed in real wastewater sequencing data. Hence, we have created additional benchmarking datasets and performed additional computational experiments to evaluate the impact on our lineage abundance predictions. For full details and results please see the supplementary material, section 3.2. These experiments show that amplicon dropout and extreme variability in coverage has relatively little impact on the accuracy of our predictions.

5) How were sequencing errors included in the benchmark? What ART model was used to simulate sequencing errors?

The ART HiSeq 2500 model was used to simulate sequencing errors. We have also looked into the impact of sequencing error rates. In this analysis we distinguish between substitution errors, insertion errors, and deletion errors.. It turns out that insertion and deletion errors have very little impact on prediction accuracy. Substitution errors, however, do impact predictions. The explanation for this is simple: due to high diversity between lineages, a substitution error will often match another lineage, thus leading to misclassification of the corresponding read and hence underestimation of the corresponding lineage. However, we also observe that at regular sequencing error rates (<1%) the impact of substitution errors on prediction accuracy remains very limited. Please see the supplementary material, section 4, for a detailed description and results for these experiments.

6) How would the workflow perform on data sets including variants not part of the manually selected reference set, e.g Omicon?

This is a very relevant question, since new lineages are emerging frequently. When a lineage is not part of our reference set, it cannot be quantified. Instead, if it is present in the dataset, it will be quantified as the lineage with the most similar sequence (most likely its most recent ancestor). If the new lineage is extremely different from any lineage in the reference set, it may not be quantified at all—in this case, the corresponding sequencing reads will remain unaligned and can be identified as such. This could give a hint that a new, highly divergent lineage is present, although in practice it is unlikely that there is no sufficiently similar lineage for the reads to align to. We have added this explanation also to the discussion (lines 387–391):

"While this reference set approach allows easy updating as new lineages appear, it also means that this approach cannot be used to detect new lineages, but only to near-optimally impute the mixture of known lineages most likely responsible for the observed data. If a new lineage is present in the sequencing data, it would be quantified as the lineage with most similar sequence (smallest edit distance), which is likely the most recent ancestor."

7) How would the workflow perform for known variants carrying additional changes not included in the reference collection?

This is not an issue; in fact, this is also the case for our benchmarking data, since the sequences that we simulate reads from are not present in our reference set (lines 492–493). Additional changes will not affect predictions, unless these changes make the genome for a given lineage locally identical to another lineage. In this case we may see slight underestimation for the original lineage, and overestimation of the alternative lineage.

The authors write that there is overestimation of low abundant and underestimation for higher abundant variants (Lines 264-269). While this can be expected, I would have wished for a reference-based method to have less problems of this sort. The authors write that "it would be interesting to evaluate predictions on real data more thoroughly, e.g., by comparing to qPCR-based variant abundance estimates per variant." This benchmark should have been performed.

We have added ddPCR-based predictions of B.1.1.7 for the New Haven wastewater samples, now shown in Figure 4 and described in lines 331–340 and 519–531. We observe that, like our predicted abundances, also the ddPCR-based abundances are highly variable. At some time points these estimates are very close together, while at other time points the estimates are quite far off. To investigate this, we aligned the sequencing reads for these datasets to the reference genome and checked for evidence of the Spike deletion at amino acid positions 69-70 (which is what the ddPCR targeted, at the time a characteristic mutation for B.1.1.7). It turned out that in our sequencing data this region is not even covered, thus confirming our observation that wastewater sequencing data is highly stochastic.

"To complement the clinical data of lineage frequencies, we have analysed our wastewater samples using ddPCR to estimate the prevalence of the B.1.1.7 lineage based on the presence of the characteristic H69/V70 deletion in the Spike gene (Fig 4). We observe that ddPCR-based abundance estimates are highly variable. While at some time points the ddPCR-based abundance estimates are very close to the predictions obtained with kallisto, at other time points these estimates are quite different. We investigated these specific instances by aligning the sequencing reads to the SARS-CoV-2 reference genome (MN908947.3) and analyzing the alignments across the Spike deletion at amino acid positions 69-70 (as targeted by ddPCR). We observed that in these sequencing data sets, this specific region is not covered by sequencing reads, thus confirming our earlier observation that wastewater sequencing data is highly stochastic."

8) It is nice to see that the abundances of variants in wastewater corresponds to the "real" GISAID abundances, but it does not come as much of a surprise either and has been shown before, e.g. in 10.3201/eid2705.204410. There is also an issue considering that GISAID data, if not coming from systematic surveillance efforts and sequencing of random samples, is not representative for population level viral variant abundances, which should be discussed.

Thank you for this reference, we were not aware of this paper and found it an interesting read. However, this paper does not quantify individual lineages in wastewater samples. Instead, the authors construct a consensus genome per wastewater sample and analyze these consensus genomes. This is very different from the task we are trying to accomplish here. To the best of our knowledge, it has not been shown before that lineage quantification from wastewater corresponds roughly to GISAID abundances across the US. We fully agree that the GISAID data is not necessarily representative of population level viral variant abundances, but for our analysis of wastewater samples from across the US it is the only information that we can possibly compare to. We discuss this issue in the Results (lines 365–367) and Discussion (lines 408–409).

Additional comments:

It is not entirely clear what FPR and FNR refers to in Table 1. It seems to be the prediction of non-existent strains or the non-prediction of existent strains, but it is not explained anywhere.

Thank you for pointing this out, your assumption is correct. We have added definitions for false positives and false negatives in the text (lines 244–247) and define FPR and FNR in the caption of Table 1.

There seems to be a correlation between Ct value and genome coverage, but there is no correlation coefficient (lines 234-236). It is of no surprise that there are less genomes with >20x coverage if the number of aligned reads is low (Figure 3b).

We agree that these findings are not surprising, which is exactly why these results confirm that wastewater sequencing makes sense and that we can filter for samples of sufficient quality. Since wastewater sequencing is a relatively new technique, we believe that sharing these findings is important to many readers.

Since some variants are present with up to 17 sequences and others with only a single sequence, it would make sense to include a test data set which shows that there is no bias towards the variants overrepresented in the reference database.

Excellent suggestion, we have performed such experiments. Our conclusion is that there is no bias towards variants that are overrepresented in the reference database. Please see the Supplementary material, section 3.3, for full details and results.

The comparison shown in Figure 5 is interesting, but stating that the variant abundances match the expected pattern seems not entirely evident, even though the authors state that "individual samples are unreliable": Particularly for B.1.429, there is no overlap between Wastewater and GISAID and even the trend for the others does not match really well.

Thank you for pointing this out, we have rephrased this a bit. What we meant with expected patterns here was not just the GISAID abundances, but also more generally the fact that B.1.427 and B.1.429 are variants which were primarily observed in California—which is exactly what we see in our predictions. Similarly, B.1.526 is a variant which was primarily observed in New York and Connecticut, which is again what we see in our predictions. Finally, B.1.1.7 is a variant that was observed across the US, but at the time that these samples were taken it was most abundant in Florida, which is also what our predictions indicate. We have further clarified this in the manuscript (lines 367–376):

"We observe that, while individual samples are unreliable, the predicted lineage abundances match expected patterns across the US from the times of sampling: B.1.1.7 was predicted most abundantly in Florida; B.1.427 and B.1.429 were primarily found in California; and B.1.526 was predicted most abundantly in New York and Connecticut. Other lineages (B.1.351, P.1) were not observed in GISAID for these states at the time of sampling and our predictions for these lineages agree: B.1.351 was predicted to be present at very low frequency in 4 samples and absent in all other samples; P.1 was predicted present in a single dataset at 1% abundance and absent in all others (Fig S14, Additional File 1). Although these predictions may be false positives, at the time P1 was thought to be likely at such low prevalence that these cases were not picked up by the sequencing efforts in place."

In addition to what is there the Github repository should include a requirements.txt or something similar for ease of installation. Right now every single software has to be installed manually via trial and error. After this is done, the example runs and produces the expected result.

We have added a dependencies.txt file to the GitHub repository with the specific conda packages used in our analysis. This allows users to immediately install a complete conda environment to run the pipeline.

Additionally, problems specific for viral abundance estimation from RNA-seq data as e.g. described here <https://journals.asm.org/doi/pdf/10.1128/JVI.01342-18> are not mentioned.

The wastewater sequencing data that we generated here was not via traditional RNA-seq, rather by amplicon-based virus sequencing. However, some of the limitations of RNA-seq data

still apply here, such as the pros and cons of short- and long-read sequencing. More directly applicable to this study would be how primer mismatches can bias lineage frequency estimates. We added the following to the limitations section of the discussion:

"Amplicon-based sequencing, as employed here, can also bias the detection of within-sample lineage frequencies if there are mismatches to the primer sequences. One of the strengths of using transcriptome analysis tools on such data is that we only analyze lineage abundance: as long as the amplicon biases even out across the genome, there will be little impact on prediction accuracy. However, the main limitation of this approach is that any amplification bias goes unnoticed, as we do not consider mutation frequencies of individual sites."

Second round of review

Reviewer 1

All my comments were properly addressed

My only issue is how the comments regarding the viral quasispecies was addressed

"Per consequence, regular viral quasispecies quantification approaches (see Eliseev et al.20 for an overview) are not suitable for this type of data---instead, we need an approach that is robust to the variability in wastewater sequencing data quality."

There is no evidence to suggest this statement. Instead a systematic and rigorous benchmarking involving real data sets needs to be performed to evaluate if quasispecies are suitable or not

I recommend discuss this in the Discussion section

Response to reviewer #1

Comment:

All my comments were properly addressed

My only issue is how the comments regarding the viral quasispecies was addressed

"Per consequence, regular viral quasispecies quantification approaches (see Eliseev et al.20 for an overview) are not suitable for this type of data---instead, we need an approach that is robust to the variability in wastewater sequencing data quality."

There is no evidence to suggest this statement. Instead a systematics and rigorous benchmarking involving real data sets needs to be performed to evaluate if quasispecies are suitable or not. I recommend discuss this in the Discussion section.

Response:

Thank you for carefully reading our revised manuscript. While we understand the interest in the use of viral quasispecies analysis tools in the context of wastewater analysis, we would like to clarify that there are essential differences between these tools and the lineage abundance quantification problem that we address. Most importantly, viral quasispecies analysis tools do not quantify a set of pre-defined lineages/variants, they solve a different problem (namely reconstruction of all sequences present, without taking lineage definitions into account). Moreover, these tools would be severely hampered by the low quality of wastewater sequencing data, which is very different from regular virus sequencing data. We have added further explanation of these differences to the manuscript (lines 118--120). For these reasons, we believe that applying viral quasispecies tools to wastewater sequencing data is beyond the scope of this manuscript, but could be an interesting future direction. We have followed the reviewer's suggestion and added this as a final paragraph to our discussion (lines 435--443).